# Improving continuous-flow analysis of triple oxygen isotopes in ice cores: insights from replicate measurements

Lindsey Davidge[1], Eric J. Steig[1], Andrew J. Schauer[1]

[1]Department of Earth and Space Science, University of Washington, Seattle, 98195, USA

*Correspondence to*: Lindsey Davidge (ldavidge@uw.edu)

**Abstract.** Stable water isotope measurements from polar ice cores provide high-resolution information about past hydrologic conditions and are therefore important to understanding earth's climate system. Routine high-resolution measurements of $\delta^{18}O$, $\delta D$, and deuterium excess are made by continuous-flow analysis (CFA) methods that include laser spectrometers. Cavity ring-down laser spectroscopy (CRDS) allows for simultaneous measurements of all stable water isotopes, including $\delta^{17}O$ and $^{17}O$

excess ($\Delta^{17}O$); however, the limitations of CFA methodologies for $\Delta^{17}O$ are not well understood. Here, we describe a measurement methodology for all stable water isotopes that uses a CFA system coupled to a CRDS instrument. We make repeated measurements of an ice-core section by this method to explore the reproducibility of CFA-CRDS measurements for $\Delta^{17}O$. Our data demonstrate that the CFA-CRDS method can make high-precision measurements of $\Delta^{17}O$ (<5 per meg at averaging times >3000 s). We show that the variations within our CFA ice-core measurements are well matched in magnitude

and timing by the variations within the discrete CRDS measurements; we find that calibration offsets generate most of the variability among the replicate datasets. When these offsets are accounted for, the precision of CFA-CRDS ice-core data for $\Delta^{17}O$ is as good as the precision of $\Delta^{17}O$ for continuous reference water measurements. We demonstrate that this method can detect seasonal variability in $\Delta^{17}O$ in Greenland ice, and our work suggests that the measurement resolution of CFA-CRDS is largely defined by the melt and measurement rate. We suggest that CFA-CRDS has the potential to increase measurement

resolution of $\delta^{17}O$ and $\Delta^{17}O$ in ice cores, but also highlight the importance of developing calibration strategies with attention to $\Delta^{17}O$.

## 1 Introduction

Records of water isotopologues from ice cores are fundamental to the study of past climate processes (Dansgaard, 1964).

Oxygen ($\delta^{18}O$) and hydrogen ($\delta D$) isotope ratios have been measured routinely in ice core samples and in other natural waters due to their well understood, first-order equilibrium fractionation relationship to atmospheric temperature (Jouzel et al., 1997). Additionally, deuterium excess (*d*) is commonly used as an indicator of kinetic fractionation processes within the hydrologic cycle (Merlivat and Jouzel, 1979). Deuterium excess is conventionally defined as:

$d = \delta D - 8*(\delta^{18}O)$                                                         (1)

Barkan and Luz (2005) showed that measuring $\delta^{17}O$ and $\delta^{18}O$ at sufficiently high precision allows for the determination of $^{17}O$ excess ($\Delta^{17}O$), a quantity that, like $d$, also reflects nonequilibrium fractionation processes such as sea-surface humidity (Uemura et al., 2010) and supersaturation effects during snow formation (Schoenemann et al., 2014). $\Delta^{17}O$ is defined by Luz and Barkan (2010) as the deviation in $\delta^{17}O$ from the global meteoric water line:

$\Delta^{17}O = \ln(\delta^{17}O+1) - 0.528 \ln(\delta^{18}O+1)$                                    (2)

where $\delta$ ("delta") values are expressed as a unitless fractional deviation from Vienna Standard Mean Ocean Water (VSMOW; see e.g., Schoenemann et al., 2013, for a complete discussion of nomenclature).

Measurements of $\delta^{18}O$, $\delta D$, and $d$ by laser spectroscopy have been demonstrated by many laboratories (e.g., Kerstel et al., 1999; Iannone, 2010; Steen-Larsen, 2014; Schauer et al., 2016; Jones et al., 2017a); for water-isotope measurements of ice

cores, it is increasingly common to couple a laser spectrometer with a continuous-flow analysis (CFA) system. CFA processing reduces sample handling and can produce very high depth-resolution (originally described by Gkinis et al., 2010; 2011). Highly resolved water isotope measurements are advantageous for a variety of studies, such as those that use the water-isotope diffusion length to infer information about firn processes or to reconstruct temperature histories (e.g., Gkinis et al., 2014; Kahle et al., 2018; 2021; Jones et al, 2017b). It is desirable to obtain measurements of $\delta^{17}O$ and $\Delta^{17}O$ at a resolution comparable to

that for $\delta^{18}O$, $\delta D$, and $d$. Corresponding measurements of both $\Delta^{17}O$ and $d$ – which have differing sensitivities to kinetic fractionation processes – could help to disentangle the various processes that influence water isotope values during evaporation, atmospheric transportation, and snow formation (Angert et al., 2004; Uemura et al., 2010). However, measurements of $\Delta^{17}O$ require much higher precision than the other water-isotope ratios and have therefore generally been obtained by isotope-ratio mass spectrometry (IRMS) (Luz and Barkan, 2010; Landais et al., 2008; 2012; Schoenemann et al.,

2013; 2014). Because the IRMS method is relatively expensive and time-consuming, $\Delta^{17}O$ measurements from ice cores are limited in spatial and temporal resolution (e.g., Landais et al., 2008; Schoenemann et al., 2014; Aron et al., 2021). CFA for $\Delta^{17}O$ has the potential to address this limitation.

Laser spectroscopy enables simultaneous measurements of $\delta^{17}O$, $\delta^{18}O$, and $\delta D$ (and therefore $d$ and $\Delta^{17}O$). Steig et al. (2014) developed a cavity-ring-down laser spectrometer (CRDS) for $\Delta^{17}O$ analysis, sold commercially as the Picarro L2140-$i$; other

instruments with different spectroscopic methods have also been developed for $\Delta^{17}O$ analysis (e.g., Berman et al., 2013; Tian et al., 2016). Schauer et al. (2016) demonstrated that the L2140-$i$ CRDS configured with an autosampler can routinely measure $\Delta^{17}O$ from discrete water samples with precision and accuracy comparable to IRMS methods. Steig et al. (2021) obtained continuous measurements of all water isotope quantities ($\delta^{17}O$, $\delta^{18}O$, $\delta D$, $d$, $\Delta^{17}O$) on an ice core from the South Pole by using

the L2140-*i* CRDS coupled to the CFA system developed by Jones et al. (2017a). However, despite the potential shown by these studies, the adoption of CFA-CRDS for $\Delta^{17}O$ faces two primary challenges. First, the integration time required for high-precision $\Delta^{17}O$ measurements by CRDS – approximately 1000 s to achieve precision of 10 per meg (Steig et al., 2014) – is much greater than the integration time required to achieve meaningful precision for $\delta^{18}O$, $\delta D$ or $d$. Second, the CFA system – i.e., the melting and vaporization process used to introduce an ice core sample into the CRDS – may further degrade the measurement quality by processes that are not yet well understood. For example, Steig et al. (2021) identified occasional large (>20 per meg) offsets in CFA-CRDS $\Delta^{17}O$ in their measurements of the South Pole ice core; the cause of these offsets was unclear. It is our goal to characterize the reproducibility of replicate ice core measurements of $\Delta^{17}O$ by CFA-CRDS.

Here, we describe a CFA-CRDS measurement methodology that was designed for high-resolution measurements of $\Delta^{17}O$. We take advantage of archived ice core samples from Summit, Greenland to make repeated CFA-CRDS measurements of $\Delta^{17}O$. These samples (collected by Hastings et al., 2009) provide an opportunity to explore the potential and limitations of $\Delta^{17}O$ measurements by CFA-CRDS more fully. We use replicate measurements made by CFA-CRDS and discrete CRDS methods to assess the reproducibility of CFA-CRDS $\Delta^{17}O$ data and to identify sources of measurement error.

## 2 CFA-CRDS design and configuration

We use a CFA processing line in combination with a CRDS laser spectrometer (L2140-*i*, Picarro Inc., as in Steig et al., 2014) to measure $\Delta^{17}O$ of ice core samples. The function of the CFA line is to generate a continuous supply of constant-humidity sample vapor to the CRDS analyzer; to achieve this, we have built a custom vaporizer unit that is described below. A constant stream of vaporized sample is important because errors in isotope-ratio measurements can arise from inconsistent vapor pressure at the CRDS inlet (Gkinis et al., 2011; Schauer et al., 2016). Finally, we aim to reduce diffusion and mixing within the CFA system to avoid smoothing the resulting measurements.

### 2.1 Custom vaporizer design

Continuous and complete vaporization is critical to reducing errors in all CRDS stable water isotope measurements, and it is especially important for attaining the per-meg precision necessary to detect meaningful variations in $\Delta^{17}O$. Previous studies have achieved continuous vaporization by heating sample water in the presence of dry air, either within an insulated stainless-steel tee (e.g., Gkinis et al., 2010; 2011) or within a concentric glass nebulizer with a vaporizing tube (e.g., Emanuelsson et al., 2015; Jones et al., 2017a). Gkinis et al. (2010; 2011) designed a flash vaporization process to instantaneously vaporize a continuous stream of sample water; the flash vaporization process involves a continuous stream of water that is combined with a continuous stream of dry air inside a 0.50 mm internal diameter stainless-steel tee that is maintained at near-ambient pressure. Steig et al. (2021) measured $\Delta^{17}O$ by CFA-CRDS with the CFA configuration of Jones et al. (2017a): a continuous stream of water sample at 1030 kPa (150 psi) is aerosolized within a concentric glass nebulizer; the aerosolized sample droplets then

evaporate completely within a 1.8-cm internal diameter, 20 cm long glass vaporizing tube that is heated to 200 °C. In this configuration, the CRDS analyzer draws vaporized sample from the vaporizing tube, and excess sample vapor is vented to laboratory air (Jones et al., 2017a; Steig et al., 2021). Two critical differences between the Gkinis et al. (2010; 2011) and Jones et al. (2017a) methods are the volume of the vaporization chamber and the volume of vapor that is generated. The smaller volume of the flash vaporizer should limit signal smoothing between the vaporizer and the analyzer. However, the flash vaporization method described by Gkinis (2010; 2011) generates vapor at approximately the rate that it is required by the analyzer, whereas the nebulizer method of Jones et al. (2017a) produces an excess of vapor that is vented prior to reaching the analyzer. Producing excess vapor is another way to limit the signal smoothing upstream of the vapor vent because it increases the velocity of sample through the system.

For this study, we built a custom vaporizer unit that benefits from both the small volume of the flash vaporizer and also from the production of excess sample vapor; we also adopted additional monitoring techniques to ensure that there are stable flow conditions within the system during analysis. We use a 0.50 mm stainless-steel tee as used by Gkinis et al. (2010; 2011), but we operate our vaporizer at a much higher mixing pressure (typically 200 kPa) than is used by Gkinis et al. to produce and vent approximately thirty times more vapor than is required for analysis. A small system volume combined with a high volumetric flow rate leads to a short retention time within the vaporizer that limits mixing of adjacent ice core layers. An additional benefit of the small vaporizer volume is that flow inconsistencies (i.e., changes in sample flow rate caused by flow obstructions or bubble interruptions) that may occur within the vaporizer can be observed by the 1 Hz CRDS measurement values; patterns in water vapor concentration or instantaneous isotope readings provide information about vaporization conditions that is important for identifying and avoiding water isotope fractionation. We use CRDS observations of water concentration and uncalibrated water isotope values as well as electronic pressure sensors to infer vaporization conditions that may affect $\Delta^{17}O$. This information is used to tune the CFA-CRDS system prior to analysis, with the goal of reducing possible isotope fractionation that may cause errors in $\Delta^{17}O$; this process is described more fully in Sect. 3.1.

## 2.2 CFA-CRDS system configuration

The CFA process from the ice-core melter to the vaporizer and vapor analyzer is described below and illustrated in Figure 1. Glacial ice is melted on a 30 mm × 30 mm aluminium melt head that is fitted with four resistance heater cartridges and held at constant temperature by a PID controller (Bigler et al., 2011). Sample melt is drawn away from the melt head and through an automated selector valve (VICI, p/n C25Z-3186EMH) by a dedicated peristaltic pump, PUMP-1 (MasterFLex L/S 7535-04). The automated valve is configured to select a rotating sequence of calibration standards when ice cores are not being measured. Sample melt is carried by 0.5 mm internal diameter PFA conveyance tubing between all system components prior to the vaporizer; PFA tubing was chosen because its transparency is advantageous for identifying bubbles and investigating flow instability issues. From PUMP-1, water flows through a Darwin Microfluidics gas-permeable membrane bubble trap (44 µL internal volume, p/n LVF-3526) where bubbles are removed and vented to the laboratory air. Excess water pressure is

relieved at a vent. Sample water is drawn away from the vent by PUMP-2 (same model as PUMP-1), whose flow rate is set to match the demand of the downstream vaporizer. The vent accommodates the differences between PUMP-1, which controls the melt rate, and PUMP-2, which controls the vaporization rate. Water flows through 2 μm and 1 μm in-line filters in series to restrict the flow of particulates into the vaporizer. PUMP-2 is also preceded and followed by electronic pressure sensors PI-1 (Elveflow PS3-Small) and PI-2 (Elveflow PS4-Small) to monitor injection pressure conditions and pump and filter performance. Typically, the pressurized dry air entering the vaporizer adds backpressure on the liquid sample injection line, which damps the cyclic pressure fluctuations of the peristaltic pump and leads to less variable flow into the vaporizer. The system also includes a flow valve (FV-1) that can be used to adjust the backpressure on PUMP-2 before making a measurement.

At the vaporizer, filtered sample water is mixed and heated with dry air to produce a constant-humidity stream of vaporized sample. Immediately before entering the vaporizer, the liquid sample line is reduced to a 100 μm fused silica capillary tube. The 100 μm capillary provides sufficient flow restriction that is important for efficient vaporization while also performing well for periods of several days without clogging. The custom vaporizer includes a 0.50 mm internal diameter tee heated to 170 ºC using a PID-controlled resistance heater cartridge, similar to Gkinis et al. (2010; 2011). The vaporizer combines pressurized dry air with liquid sample, and it is set within an aluminium enclosure that is lined with 3.175 cm of calcium silicate insulation. After the sample is vaporized, the vapor is drawn into the optical cavity where it is measured, and excess vapor is vented into the laboratory. Vapor is carried from the vaporizer to the optical cavity within insulated tubing to prevent condensation.

**2.3 Design choices to mitigate memory effects**

Finally, several design choices for the CFA system are intended to reduce and characterize the memory between measurements. Because our automated selector valve is positioned immediately after the ice core melt head, reference waters pass through all components of the sample handling system except the melt head and its tubing; by design, the mixing length expected between measured ice core layers with differing isotopic composition can be approximated by the mixing length represented by transitions in reference waters if all other system conditions are identical. Mixing length within the system is reduced by increasing the flow velocity and therefore limiting the sample retention time in two ways: overall sample handling system volume and tubing diameters are minimized where possible, and excess sample volume is drawn through the entire system during CFA analysis. During ice core measurements, approximately six times more water is handled by PUMP-1 than is sent by PUMP-2 into the vaporizer; excess liquid volume is vented before PUMP-2. Similarly, approximately thirty times more vapor is generated than is analyzed; excess vapor is driven by the differential between PUMP-2 and the L2140-*i* inlet pump, and it is vented to laboratory air immediately before vapor enters the optical cavity. In this way, the liquid and vapor tubing is flushed with many times more sample volume than is required for analysis.

## 3 CFA-CRDS operations and measurements

We designed an operational sequence for reference water and ice core measurements during a period when lab work was intermittent due to the COVID-19 pandemic. The CFA system was configured to automatically measure an alternating sequence of three in-house reference waters over a period of approximately seven weeks; reference waters included Seattle tap water (SW2), West Antarctic Ice Sheet Divide snow (CW), and South Pole snow (SPS2), as shown in Table 1 and indicated in Fig. 2. Measuring reference waters continuously allows us to explore the long-term changes in system calibration while also informing maintenance requirements over long timescales. When available, an operator prepared and measured an ice core section between reference water measurements. The need for frequent calibration of CRDS data for $\Delta^{17}O$ has been well documented (e.g., Schauer et al., 2016), and continuous reference water measurements ensured that there was calibration data available adjacent in time to each intermittent ice core analysis.

We operated the CFA-CRDS system to measure nine repeated sections of an ice core and a repeated sequence of internal reference waters that we used to calibrate the ice core measurements; the sequence of ice-core measurements is listed in Table 2. We also measured a replicate ice core section by discrete CRDS for comparison. Repeated reference water measurements are used to develop a calibration for the ice core data. We compare our calibrated CFA-CRDS $\Delta^{17}O$ data with the discrete measurements to evaluate this method.

### 3.1 Operational considerations to maintain efficient vaporization

Because the vaporizer is sensitive to small fluctuations in sample flow rate, a careful balance of system pressures is required to control sample flow (Gkinis et al., 2010; 2011); specifically, the pressure of the sample at the vaporizer inlet must be slightly greater than the pressure of the dry air within the vaporizer. Maintaining a balance between the air pressure and sample pressure within the vaporizer requires knowledge of both pressure conditions. We monitor pressures at PI-1 and PI-2 so that it is possible to diagnose the source of system pressure changes when they occur; we also fix the pressure of the dry air line with the backpressure regulator (typically 200 kPa). Vacuum conditions at PI-1 indicate particulate loading across the filter screen at F-1; the filter screen will clog over time and, if the filter screen is not replaced, suction from the inlet of PUMP-1 can draw a vacuum at PI-1. Vacuum conditions at PI-1 can impact the downstream peristaltic pump (PUMP-2) performance, ultimately causing inconsistent flow into the vaporizer and analyzer. Under optimal analysis conditions, the pressure is near ambient at PI-1. A decrease in pressure at PI-2 indicates upstream vacuum conditions or worn peristaltic pump tubing at PUMP-2. An increase in pressure at PI-2 indicates clogging downstream, which can occur as particulate loading within F-2, or as mineral precipitation within the capillary or vaporizer. The pressure at PI-2 generally varied between 200 kPa and 400 kPa, depending on the injection air pressure and the precipitate levels within the vaporizer or capillary tubing. High-pressure vaporizer conditions allow sample to flow despite the inevitable accumulation of precipitate within the vaporizer, which enables the system to operate in balance for days or even weeks. However, over time, precipitate accumulation within the vaporizer can

restrict the flow of air, water, or both; this typically requires re-balancing of system flow conditions, but it can occasionally require removing and cleaning vaporizer fittings with soap, water, and physical agitation.

During operation of the CFA-CRDS, intermittent reductions in water vapor concentration can occur within the vaporizer, which can produce perturbations in the isotope data. Gkinis et al. (2010; 2011) described sample flow inconsistencies at their CFA flash vaporizer that cause extreme outliers in isotope data, though the cause of the fluctuations was unclear. We observe similar fluctuations, and the pressure sensor data provide insight into their cause. We find that the most common causes of such variations are microbubbles entering the vaporizer owing to particulate loading, which can cause poor debubbler performance and can also cause blockages to form within small tubing fittings. Microbubbles that remain suspended in the fluid stream after the debubbler cause volumetric flow rate reductions at the vaporizer inlet. Blockages within fittings upstream of PUMP-2 can cause extreme vacuum conditions before the pump (i.e., pressure observations associated with blockages were as low as -140 kPa before PUMP-2 instead of the typical ambient conditions); this can lead to the contamination of system tubing with small bubbles that also cause temporary flow reductions. To avoid these inconsistencies, we find that it is important to periodically clean the de-bubbler unit and to maintain ambient pressure at the PUMP-2 inlet by replacing clogged filter screens or tubing fittings. Although data outliers could be systematically removed (as done in Gkinis et al., (2011)), occasional bubbles do not substantially impact the isotopic mean value of our ice core measurements and are retained here. We do exclude some reference-water calibration data, where bubble interruptions are most frequent due to limited operator oversight during the automated reference water measurements. Calibration measurement criteria are discussed in Sect. 3.4, below.

In addition to monitoring pressure evolution across the system, we can also observe the quality of vapor at the CRDS analyzer via characteristic patterns that arise in the CRDS data. Specific patterns in water vapor concentration and $\delta^{18}O$ that emerge from unstable flow into the vaporizer are shown in Fig. 3. We observe that pulsating flow conditions can cause incomplete vaporization, identified by anticorrelated fluctuations in water vapor concentration and $\delta^{18}O$. When a pulse of water overwhelms the vaporizer, the isotopic composition becomes lighter as $H_2^{16}O$ preferentially evaporates from the flow steam; as the vaporizer dries out between pulses, the isotopic composition becomes heavier, exhibiting an evaporation signal. Pulsating flow conditions are caused by pressure fluctuations from the peristaltic pump (>70 kPa) when insufficient backpressure is applied on PUMP-2. The resulting patterns have a large amplitude (up to 10,000 ppm for water vapor, and several ‰ for $\delta^{18}O$) and a frequency that mirrors that of the peristaltic pump (e.g., Fig. 3a). The observed fractionation that occurs during these vaporization conditions leads to large calibration bias for $\Delta^{17}O$, causing errors of tens to hundreds of per meg. If there is sufficient backpressure at PUMP-2, the pressure readings at PI-2 are typically <40 kPa. We attribute small fluctuations in $\delta^{18}O$ that are anticorrelated with water vapor concentration to the incomplete vaporization of individual droplets (e.g., Fig. 3b). Because inconsistent flow into the vaporizer can cause isotope fractionation and because it is important to measure calibration standards under the same conditions as the ice core samples, we tune the system to maintain steady pressure

readings at the vaporizer inlet prior to calibration standard and ice core analysis, as discussed in Sect. 3.2.; vapor concentration data that are typical of a well-maintained CFA system are shown in Fig. 3d.

## 3.2 Measuring $\Delta^{17}O$ by CFA-CRDS in ice core samples

Approximately twelve to twenty-four hours before making an ice core measurement, an operator maintained the CFA system to balance the flow rate into the vaporizer. For example, when indicated by anomalously high or low pressure sensor data, the filter screens, peristaltic pump tubing, or capillary tubing were replaced. When indicated by CRDS data trends as in Fig. 3, the vaporizer components were cleaned. Returning the CFA system to a balanced state before making measurements of all reference waters increases the likelihood of having usable, high-quality calibration data against which to calibrate the ice core
samples. At other times when ice core measurements were not made, the system occasionally drifted out of balance and was not actively maintained, such that some of the reference water measurements are of lower quality than those used to calibrate the ice core measurements. This is discussed in more detail in Sect. 3.4.

We cut an 87.5 cm ice core sample, from ~92 m depth beneath the surface at Summit, Greenland, into nine 26 mm square slices to prepare them for continuous analysis. After preparing these nine CFA sticks, a tenth section of core was cut into 63
discrete depth intervals. Discrete ice samples were melted in sealed polyethylene sample bottles in a refrigerator at 4° C. We measured the 87.5 cm section of ice ten times: the nine replicate slices were measured by the CFA-CRDS configuration described above, and the tenth measurement was made by discrete injection of 63 melt samples from the core using the commercially available vaporizer unit (Picarro p/n A0211) and automated injections as in Schauer et al. (2016). The depth resolution of the discretely measured ice is 1.39 cm.

For all CFA measurements, we made visual observations of the core height to monitor the melt rate during analysis, then later assigned a high-resolution depth equivalent for each analysis time that is based on the value of $\delta^{18}O$ and the measured depth of discrete samples. Previous work has monitored core depth with electronic distance meters (e.g., Bigler et al., 2011; Jones et al., 2017a), and such measurements are critical for depth registration for routine CFA measurement campaigns. Here, we forego electronic depth registration and instead adjust initial depth estimates for each core section by aligning the seasonal
cycle of $\delta^{18}O$ for all core samples. Summit, Greenland has a modern annual accumulation rate of 24 ± 5 cm (ice equivalent) per year (Meese et al., 1994; Dibb and Fahnestock, 2004; Hawley et al., 2008; 2021), and we expect to see two to three years represented by the core sample that we measured in replicate (Hastings et al., 2009). Assigning depths by aligning the $\delta^{18}O$ variations should largely eliminate depth-registration errors, since the strong seasonal $\delta^{18}O$ variations must be essentially identical in each replicate sample, and the signal to noise ratio for $\delta^{18}O$ is very high. We compressed the depth scale of each
CFA record to maximize the cross-correlation of $\delta^{18}O$ ($0.93<r<0.99$) between the CFA measurements and the discrete measurements. We then assigned each CFA analysis time a depth equivalent based on the depth of the corresponding discrete $\delta^{18}O$ data. We note that the amplitude of the seasonal variations in $\delta^{18}O$ is somewhat compressed in the lower ~30 cm of this

core sample, so the depth designations for this interval are likely a greater source of error than in the rest of the ice. Nevertheless, we are confident that our depth registration is precise to within a cm or better, determined by assessing the variance in depth assignments at inflection points.

## 3.3 Operational choices to mitigate memory effects

Mitigating memory effects is important for both ice-core and reference water measurements; in addition to the design choices highlighted in Sect. 2.3, there are several operational choices that were made to reduce the memory between isotopically distinct waters. For example, increasing the pump rates at PUMP-1 during ice core analysis should drive shorter retention times within the tubing upstream of the liquid vent, which should reduce system mixing. In this way, the transition times for reference waters (shown in Fig. 2) are a conservative estimate of mixing effects. The transition time between measurements of reference waters generally varied between 180 and 360 s. We therefore assume a conservative mixing time of 360 s during reference water transitions, and we ignore the 360 s that initiate and conclude each reference water measurement. Before measuring each section of ice (which is typically ~1 m long), we also condition the system with at least ten minutes of water with similar isotopic composition to prevent mixing between isotopically disparate reference waters and ice core samples at the beginning of the analysis. Finally, the replicate CFA-CRDS measurements that are the focus of this study provide a practical evaluation of the effects of memory on measurement fidelity in this configuration.

## 3.4 Calibrating CFA-CRDS $\Delta^{17}O$ data

To achieve an accurate calibration, similar treatment of reference waters and sample melt during vaporization is critical. For this study, we measured the calibration standards immediately before and after measuring an ice core section; this ensures the most comparable treatment of reference waters and sample melt. Achieving similar treatment also requires that the system is stable during the entire measurement period, including reference water measurements and ice sample measurements. Because an individual ice core measurement takes a few hours at the melt rates that we employ, we limit our reference water measurements to three hours each to increase the likelihood that the complete sequence of reference waters and ice core samples is measured under similar CFA and CRDS conditions.

To calibrate our $\Delta^{17}O$ measurements, we create a two-point linear calibration for $\delta^{17}O$ and $\delta^{18}O$ from the nearest measurements of our internal reference waters, SW2 and SPS2; a third reference water (CW) is used as an independent verification of the calibration. The values of SW2, CW, and SPS2 have been measured independently and are normalized to the VSMOW-SLAP scale as in Schoenemann et al. (2013). Three-hour measurements of these alternating reference waters were made between measurements of the ice core sample; to measure a reference water, the selector valve was electronically switched to the next sequential water, causing the peristaltic pump to draw reference water volume from a new standard container. Reference water measurements were automated and typically unsupervised. Because measurement conditions evolve over time due to particulate loading and mineral precipitation within the CFA components and because there were periods of time during the

analysis window when no operator was available to monitor system conditions, there were periods of time during which the water vapor concentration was outside the ideal range, during which large bubbles or other flow inconsistencies degraded the quality of reference water data, or during which the CFA system was not operating; consequently, only about 50% of the analysis time captured by the study period is included in this analysis, as described below and in Table 2. We automatically reject calibration data and measurements of CW that were generated from water vapor concentration beyond the targeted range (i.e., <20,000 or >50,000 ppm) or data with insufficient vaporizer operations, indicated by $\sigma_{\delta18O}$ > 0.5 ‰ across the measurement window. Typical variability of water vapor concentration within a single 3 hour period is 0.5% to 5%. We identify transitions from one reference water to the next in the data by the second derivative of $\delta D$, and assign known standard values based on the uncalibrated measurement values of $\delta D$. We include measurements of SW2 and SPS2 that contain at least 6000 s of analysis time and we trim 360 s of data from the beginning and end of each measurement interval to avoid memory effects. The mean and standard deviation of the analysis time for calibration standard data is 9350 s ± 660 s. To calibrate all ice-core and CW measurements made during this study, we use 47 continuous, 3 hour measurements of SW2 and 40 continuous, 3 hour measurements of SPS2. All analyses include measurements for $\delta^{17}O$, $\delta^{18}O$, and $\delta D$. Calculations of $d$ and $\Delta^{17}O$ were obtained from the calibrated $\delta$ values as given in Eqs. (1) and (2), respectively. Calibration for $\Delta^{17}O$ is more completely described below.

The calibration data used for all measurements is generated from adjacent measurements of SW2 and SPS2 that meet the screening criteria above; calibration data and the sequence of CFA-CRDS measurements are provided in Table 2. For each calibration of CW or ice core data, we employ the nearest measurements of SW2 and SPS2 for the calibration. The calibration is performed separately for $\delta^{17}O$ and $\delta^{18}O$: using a least-squares approach, we fit a linear equation to the uncalibrated average measurements so that the calibrated SW2 and SPS2 measurements match their known values. The calibration equation therefore becomes:

$$\delta_{calibrated} = m^* \delta_{uncalibrated} + b, \tag{3}$$

where $\delta$ represents either $\delta^{17}O$ or $\delta^{18}O$. An account of m and b for both $\delta^{17}O$ and $\delta^{18}O$ is shown for all measurements in Table 2. Finally, $\Delta^{17}O$ is calculated from the calibrated values of $\delta^{17}O$ and $\delta^{18}O$:

$$\Delta^{17}O_{calibrated} = \ln (\delta^{17}O_{calibrated} +1) - 0.528 \ln (\delta^{18}O_{calibrated} + 1). \tag{4}$$

The mean and standard deviation of all CW measurements of $\Delta^{17}O$ during the analysis period is 25±12 per meg (n=53). The subset of CW measurements with the most consistent CFA operations – and therefore the lowest variability for $\delta^{18}O$ ($\sigma$ < 0.06 ‰) – had corresponding $\Delta^{17}O$ values of 25±6 per meg (n=36). Low variability among reference water measurements gives confidence in the use of this system for this study of replicate ice-core measurements.

## 3.5 Processing CFA-CRDS $\Delta^{17}O$ data

After assigning approximate depth values and calibrating the ~1 Hz data, we discretize the CFA-CRDS data by binning the
calibrated data into prescribed depth intervals and averaging across the entire interval. This enables a direct comparison
between the continuous CFA-CRDS timeseries and the discrete CRDS measurements. Small differences in the instantaneous
melt rate cause some variability in the data-averaging duration for each reported measurement; the typical instantaneous melt
rate was ~0.3 cm/min, but rates ranged from ~0.1 cm/min to ~0.4 cm/min during analysis. We report our CFA-CRDS
measurements with 1.39 cm resolution to match the resolution of our discrete CRDS measurements. We also explore the effects
of depth resolutions that range from 0.5 cm to ~40 cm, given that increasing the averaging window of the ~1 Hz spectroscopic
measurements reduces instrumental noise (e.g., Werle et al., 1993; Gkinis et al., 2010; 2011; Steig et al., 2014; 2021; Schauer
et al., 2016; Jones et al., 2017a).

## 4 Results and analysis

Our isotope measurements capture a period of approximately two years of precipitation, as expected for a Greenland ice core
from the depth we analyzed (discussed in Sect. 3.2). The seasonal cycle of $\delta^{18}O$ is shown in Fig. 4. We estimate that our depth
assignments are accurate to <7 mm throughout the core by determining the variability in depth assignments at all inflection
points; this allows us to compare CFA-CRDS measurements of $\Delta^{17}O$ at the ~cm scale, appreciably finer resolution than has
previously been reported. Our comparison quantifies the reproducibility of our measurements and identifies sources of
variability among these CFA-CRDS $\Delta^{17}O$ data.

### 4.1 Seasonal $\Delta^{17}O$ variations in replicated CFA and discrete measurements

We compare our CFA-CRDS data for $\Delta^{17}O$ with discrete CRDS measurements to evaluate the CFA-CRDS method. We present
the mean value and standard error of all replicate measurements in Fig. 4 with 1.39 cm averaging (representing approximately
270 s of data per interval for each individual CFA-CRDS replicate); Fig. 4 also shows the discrete CRDS measurements with
the root mean square error of the corresponding discrete reference water measurements. The mean of all CFA-CRDS
measurements (representing more than 2000 s of data per interval) is well correlated with the discrete measurements ($r = 0.52$,
where $0.28 \leq r \leq 0.69$ with 95% confidence), especially in the upper 50 cm of the core ($r = 0.74$, where $0.54 \leq r \leq 0.88$ with
95% confidence). Both the CFA-CRDS data and the discrete CRDS data show clear seasonal $\Delta^{17}O$ variations at this
measurement resolution that are matched in magnitude and timing.

### 4.2 Error attribution for CFA-CRDS $\Delta^{17}O$ measurements

Next, we characterize the variability observed among our nine CFA-CRDS measurements. In addition to the depth alignment
errors discussed above, sources of variability introduced by the CFA-CRDS method may include high-frequency instrumental

noise, calibration errors, and smoothing or bias generated by mixing within the CFA system. High-frequency, high-amplitude noise (~1 ‰) in the uncalibrated CRDS data is inherent to the instrument and can cause large aberrations from the true value of $\Delta^{17}O$, especially over short averaging times; long averaging times (>1000 s) are typically used when measuring $\Delta^{17}O$ by CFA-CRDS to minimize instrumental noise. Calibration errors in $\Delta^{17}O$ occur when measurement treatment differs between calibration standards and samples or between calibration standards; this can cause fractionation to occur in the uncalibrated $\delta^{17}O$ and $\delta^{18}O$ measurements, leading to biased calibration slope and intercept information. Despite efforts to stabilize vaporizer system conditions prior to ice core sample analysis and to measure ice core samples with the same treatment, it is likely that some calibration errors persist in our ice core data because it is not possible to measure the standards and the sample at the same time. Finally, CFA measurement error for $\Delta^{17}O$ may result from mixing isotopically distinct waters during CFA processing or from other processing issues that affect the internal variability (i.e., perceived seasonality) of the continuous ice core measurement.

Typical CRDS characterization studies have used repeated measurements of reference waters to identify measurement error; for this study, we instead use repeated measurements of an ice core to characterize the sources of the measurement error. By measuring reference waters, it is possible to approximate the precision of the uncalibrated measurements by determining the effect of averaging time on the intrinsic noise of the measurement; it is also possible to quantify the variance of the calibrated, averaged data. Our best data for CW were measured at $25 \pm 6$ per meg, but without additional information, it is not straightforward to identify whether the error associated with this measurement is caused by instrumental limitations, calibration bias, or other CFA processing effects. Our replicate CFA-CRDS measurements provide an opportunity to interrogate the source of CFA-CRDS errors because we can separately analyze the variability internal to each timeseries (e.g., due to the seasonal cycle of $\Delta^{17}O$ or due to CFA errors) and the variability between the mean values for each ice-core replicate (e.g., due to calibration offsets); further, we can compare this variability with instrument expectations at different averaging times

To isolate the error imparted by the calibration strategy, we processed the data in two ways: first, we calibrated the data as described above, and second, we set the mean values of all calibrated CFA measurements equal to the mean value of the discrete CFA measurements in order to consider only the variability internal to each measurement. Steig et al. (2021) demonstrated that making a linear adjustment to the calibration intercept for $\delta^{17}O$ and $\delta^{18}O$ could reduce the noise of their CFA-CRDS measurements for $\Delta^{17}O$ in the SPC14 core. They exploited additional reference water information taken before or after the CFA measurement to define an adjusted calibration intercept value. Here, we can instead use the mean value of the CFA-CRDS measurements themselves, further eliminating uncertainty around this correction by setting the mean of each calibrated CFA-CRDS timeseries equal to the mean value of our discrete ice core measurements; in this way, we are able to eliminate offsets in calibration and examine the variability within each continuous measurement. We define the calibration-adjusted $\Delta^{17}O$ data as below:

$$\Delta^{17}O_{\text{adjusted CFA}} = \Delta^{17}O_{\text{calibrated CFA}} + \frac{1}{n} * \sum_{i=1}^{n} \Delta^{17}O_{\text{calibrated discrete}}(i) - \frac{1}{n} \sum_{i=1}^{n} \Delta^{17}O_{\text{calibrated CFA}}(i), \qquad (5)$$

where the value of n, which represents the number of data points per meter, varies as a function of the depth resolution.

The calibration offset error is therefore $\Delta^{17}O_{\text{adjusted CFA}} - \Delta^{17}O_{\text{calibrated CFA}}$. Equivalently, $\Delta^{17}O_{\text{adjusted CFA}}$ can be expressed in terms of $\delta^{17}O$ and $\delta^{18}O$, using calibration correction information that is based on the differences between average $\delta^{17}O$ and $\delta^{18}O$ values for the discrete and continuous datasets. That is,

$$\Delta^{17}O_{\text{adjusted CFA}} = \ln(m_{17} * \delta^{17}O_{\text{uncalibrated CFA}} + b_{17} + b_{17\_corr} + 1) - 0.528 \ln(m_{18} * \delta^{18}O_{\text{uncalibrated}} + b_{18} + b_{18\_corr} + 1). \qquad (6)$$

where the correction values $b_{17\_corr}$ and $b_{18\_corr}$ are defined by the difference in mean $\delta$ for CFA and discrete measurements. This calibration adjustment method is analogous to that used in Steig et al. (2021).

Evaluating both $\Delta^{17}O_{\text{calibrated CFA}}$ and $\Delta^{17}O_{\text{adjusted CFA}}$ allows us to disentangle the calibration offset error from other sources of measurement error. We discretized the CFA-CRDS data to a series of depth-resolution schemes that ranged from 1.39 cm to 43.75 cm; the data are provided for three different depth resolutions in Fig. 5. We calculated the standard error for all depth intervals across all measurement resolutions. The total error for the $\Delta^{17}O_{\text{calibrated CFA}}$ is approximated by the black line in Fig. 6, which represents the mean of the standard error across all depth intervals for all measurement replicates. The total span of the standard error at every depth interval is provided by the grey shaded region. Similarly, the blue shaded region shows the total span of the standard error for $\Delta^{17}O_{\text{adjusted CFA}}$, and the blue line is the mean error. The region between the two solid lines is the fraction of the total error that can be attributed to the calibration offset. Figures 4 and 6 indicate that the calibration offset noise is essentially indistinguishable from the instrumental noise at small averaging time, so the calibration offset adjustment does little to improve the measurement for the best resolved data. The results show that the total error is <10 per meg for all data. The total error is ~5 per meg at averaging times longer than ~3000 s, which corresponds to depth averages of ~15 cm at the melt rates we used. Figure 6 also shows that the error that arises from differences in internal variability (i.e., the CFA error) for the CFA-CRDS data is <2 per meg by ~3000 s and that the total error is dominated by calibration intercept error at long averaging times.

Finally, we directly compare the variability of our CFA-CRDS data with the variability of reference waters measured by CFA-CRDS, which is determined by an Allan variance analysis. An Allan variance analysis quantifies the relationship between signal noise and integration time (Allan, 1966; Werle et al., 1993); for CRDS data, this analysis of reference water measurements is commonly used to approximate the measurement precision of the system for any given measurement duration (Gkinis et al., 2010; Steig et al., 2014). We determine the Allan deviation (square-root of the Allan variance) from a long continuous analysis (~8.5 hours) of the SW2 reference water made during our analysis window (see Table 2); the result is shown in Fig. 7. Differences between the Allan deviation and the standard deviation of our measurements should confirm

whether the magnitude and timing of the variability is as precise as during reference water measurements, or if there are other changes imparted by the CFA system or calibration that may degrade CFA-CRDS data quality. We find the standard deviation $\sigma_{calibrated\_CFA-\Delta17O}$ among all nine $\Delta^{17}O_{calibrated\ CFA}$ datasets, averaged over integration windows that vary from 5 mm to 43.75 cm.

This analysis compares the variability of the final, calibrated measurements along the depth of the core sample with the variability of the reference water measurement, and ultimately quantifies the reproducibility of our CFA-CRDS measurements. We track the analysis time associated with each averaging interval and overlay the measured $\sigma_{CFA-\Delta17O}$ with corresponding mean integration time for each depth interval on Fig. 7a.

Figure 7a shows generally good agreement between $\sigma_{calibrated\_CFA-\Delta17O}$ and $\sigma_{Allan-\Delta17O}$ at integration times less than 400 s, but the

$\sigma_{calibrated\_CFA\Delta17O}$ data asymptotically approach a limit of 10 per meg at longer averaging times instead of following the stability trend expected by the Allan variance analysis. To evaluate to what extent this mismatch between expected and observed $\sigma$ can be attributed to errors arising from the calibration offset (as shown in Fig. 6), we repeat this analysis for the $\Delta^{17}O_{adjusted\ CFA}$ data. Figure 7b shows excellent agreement between $\sigma_{adusted\_CFA-\Delta17O}$ and $\sigma_{Allan-\Delta17O}$ at all integration times; this demonstrates that the drift in $\sigma_{calibrated\_CFA-\Delta17O}$ shown in Fig. 7a can be entirely attributed to calibration effects, and not to the CFA process

directly. Figure 7 suggests that reducing the error of calibrated CFA-CRDS measurements is not limited by the CFA process – nor to limitations of the CRDS instrument – but rather by the quality of the calibration information, which depends on the treatment and frequency of reference water measurements.

## 5 Discussion and Conclusions

### 5.1 Comparison of CFA-CRDS $\Delta^{17}O$ measurements to other $\Delta^{17}O$ measurements from Greenland

Our work complements previous studies that have examined the seasonal cycle of $\Delta^{17}O$ in polar regions, and good agreement with earlier work validates our measurements. Consistent with previous measurements from Greenland, the $\Delta^{17}O$ signal in our data is anticorrelated with $d$ and anticorrelated with the seasonal cycle in $\delta^{18}O$ (Landais et al., 2008). The measurements presented here were made from core that represents approximately two years of ice accumulation from the 1760s (Hastings et al., 2009). The measured magnitude (peak to trough) of the seasonal cycle in $\Delta^{17}O$ is ~45 per meg at 1.39 cm resolution in our

data (Fig. 4), which is in excellent agreement with the magnitude of the seasonal cycle reported previously for Greenland. Specifically, Landais et al. (2012b) reported seasonal magnitudes of ~25 per meg from a shallow firn core at NEEM (in Northwest Greenland) that represented accumulation periods between 1962-1963 and between 2003-2005; when we coarsen our measurement resolution to 3.6 cm – which approximates the ~monthly (5 cm) measurement resolution in the NEEM core (detailed in Steen-Larsen et al., 2011) – the magnitude of the seasonal cycle in our data is ~30 per meg. Low errors between

replicate values and the good agreement with previous studies strengthen confidence in the CFA-CRDS approach for high-resolution $\Delta^{17}O$.

Our results reinforce the use of the CFA-CRDS method for high-precision, high-resolution measurements of $\Delta^{17}O$ in ice cores. CFA-CRDS methods are valuable for detecting detail in $\Delta^{17}O$ variations in deep ice layers, for measuring $\Delta^{17}O$ in ice from sites with low accumulation rates, or for measuring $\Delta^{17}O$ in any glacial ice where high depth resolution is desired.

## 5.2 Addressing CFA-CRDS calibration errors in $\Delta^{17}O$

Because the error of all $\Delta^{17}O$ measurements by CRDS depends on the calibration, the importance of establishing a robust calibration strategy for CFA-CRDS cannot be understated. We iteratively revised our CFA-CRDS system and designed our calibration strategy as recommended below.

First, the CFA-CRDS configuration must be capable of stable operations that span the total duration of the ice core and reference water measurements. System stability for a given CFA system should be characterized with an Allan variance analysis. We choose to measure calibration standards immediately before and after ice core measurements to improve the likelihood of measuring the calibration standard under the same system conditions as the ice core sample. Additionally, limiting system memory and reducing the transition time between reference waters maximizes the useful fraction of reference water data, allowing measurements of longer duration or measurements of more reference waters to be made within a period of consistent CFA operations.

Next, quantifying the drift in calibration information over time can allow an operator to determine the physical controls on fractionation within a CFA-CRDS system. The change in calibration information can be used to inform system maintenance schedules or operational sequences. For example, we have observed that after operating our CFA-CRDS system for several weeks, the fractionation responses for $\delta^{17}O$ and $\delta^{18}O$ diverge, degrading the quality of calibration data for $\Delta^{17}O$. Cleaning the vaporizer fittings appears to "reset" the calibration response, suggesting that the fractionation that occurs over long timescales is a result of physical effects within the vaporizer itself owing to visible precipitate formation.

Though our system is capable of high-precision measurements for $\Delta^{17}O$, our analysis suggests that calibration bias persists in our data, which is unsurprising when considering previously published work on similar methods. The largest offsets (shown in Table 2) were associated with poor CFA stability due to a dirty vaporizer chamber. Large errors in $\Delta^{17}O$ were occasionally observed during the analysis of the South Pole ice core (SPC14); Steig et al. (2021) attributed these errors to calibration differences and performed a correction by shifting the mean value of their measurements based on the offset identified by a calibrated reference water measurement, similarly to Eq. (6). Our work supports the attribution of these errors to the calibration, and it also supports the calibration adjustment method. We recommend the use of additional reference water measurements to account for calibration offsets in $\Delta^{17}O$, and we also recommend that CFA systems are designed to ensure complete vaporization with flow conditions that are stable over long timescales. In our vaporizer, we observe that precipitate coatings can change the geometry of the vaporizer chamber and lead to incomplete vaporization over time, which degrades the quality of the calibration over time. When there is clear evidence of inconsistent vaporization (as in Fig. 3), we observe large calibration errors in $\Delta^{17}O$

by this method (tens to hundreds of per meg). Such issues likely also influence the vaporization process in other CFA systems, though they will not be readily detected in measurements of $\delta^{18}O$ or $\delta D$ if the water vapor has homogenized before reaching the analyzer.

Finally, though it is perhaps impractical to measure replicate ice core samples as we have done here, the average of our nine CFA-CRDS measurements shows that, like dual-inlet IRMS operations, stacking the CFA-CRDS data effectively averages over calibration inconsistencies. The results are comparable to highly resolved discrete CRDS or IRMS measurements. While CFA-CRDS measurements resolved to the cm scale still require long measurement times to achieve precise $\Delta^{17}O$ data (>1000 s for 10 per meg precision), stacking CFA-CRDS measurements is an effective way to increase analysis time. Typically, achieving long measurement times while maintaining high depth resolution necessitates a reduction of melt rates. In practice, reduced melt rates may be incompatible with other measurement goals (such as trace gases) during an ice core measurement campaign; reduced melt rates may also prevent the measurement of both ice core samples and calibration standards within a period of stable system operations. We show that stacking multiple CFA-CRDS measurements provides a viable alternative strategy; stacking replicate CFA-CRDS measurements improves the accuracy of the measurement by averaging over the calibration offset noise, and it improves the measurement precision or measurement resolution by increasing the total analysis time for a given depth interval.

## 6 Summary

We measured $\Delta^{17}O$ in nine replicate ice core samples using a continuous flow analysis (CFA) system combined with a cavity-ring down laser spectrometer (CRDS). We measured a tenth replicate sample by discrete CRDS methods. We show that CFA-CRDS can reliably capture cm-scale variability of $\Delta^{17}O$ in ice core samples; we measured seasonal fluctuations of ~45 per meg in $\Delta^{17}O$ from an ice core representing the preindustrial period in Greenland that agree with the discrete CRDS data and also with previously published measurements of seasonal $\Delta^{17}O$ variability in Greenland.

Our work shows that using CFA-CRDS methods can be valuable when high-precision and highly resolved measurements are desired. Our results show that mixing within the CFA system does not jeopardize CFA-CRDS measurements of $\Delta^{17}O$, even at cm-scale resolution. The mean of our stacked measurements exhibits neither a time lag nor any amplitude smoothing in comparison to the discretely prepared CRDS measurements. Rather, we show that the total error (~5 per meg for analysis times >3000 s) is dominated by calibration bias. We note the importance of developing robust calibration strategies for $\Delta^{17}O$ when making measurements by CFA-CRDS, but we demonstrate that when calibration is accounted for, CFA-CRDS for $\Delta^{17}O$ is highly reproducible and can be tailored for high-resolution and high-precision measurements.

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

**Figures and Tables**

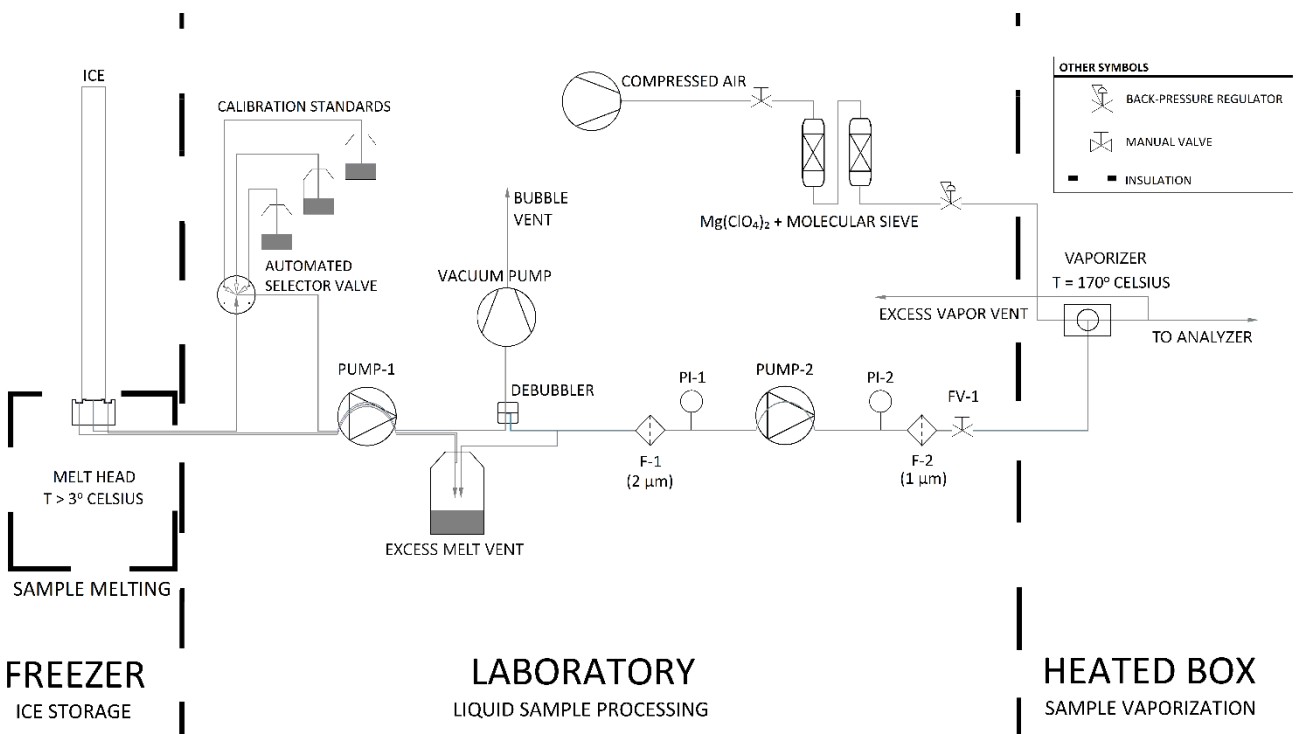

 **Figure 1: Process flow diagram for our CFA system. Thick dashed lines indicate transitions between temperature-controlled process spaces. Note that F-1 and F-2 are filters, PI-1 and PI-2 are pressure sensors, and FV-1 is a flow valve.**

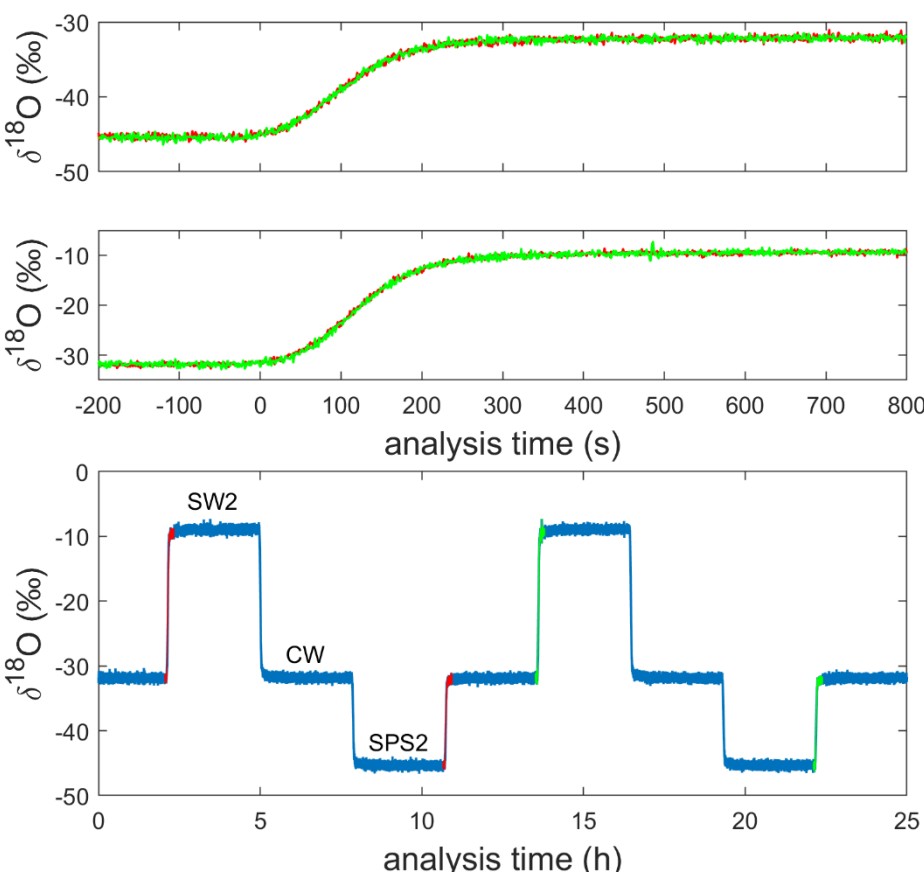

**Figure 2: Uncalibrated, 1-Hz measurements of $\delta^{18}O$ for the alternating sequence of reference waters during a full analysis day (bottom). The 200 s preceding and 800 s following four reference water transitions are shown in the other panels; two transitions (shown in orange and green) from SPS2 to CW are stacked in the top panel and two transitions from CW to SW2 are stacked in the middle panel.**

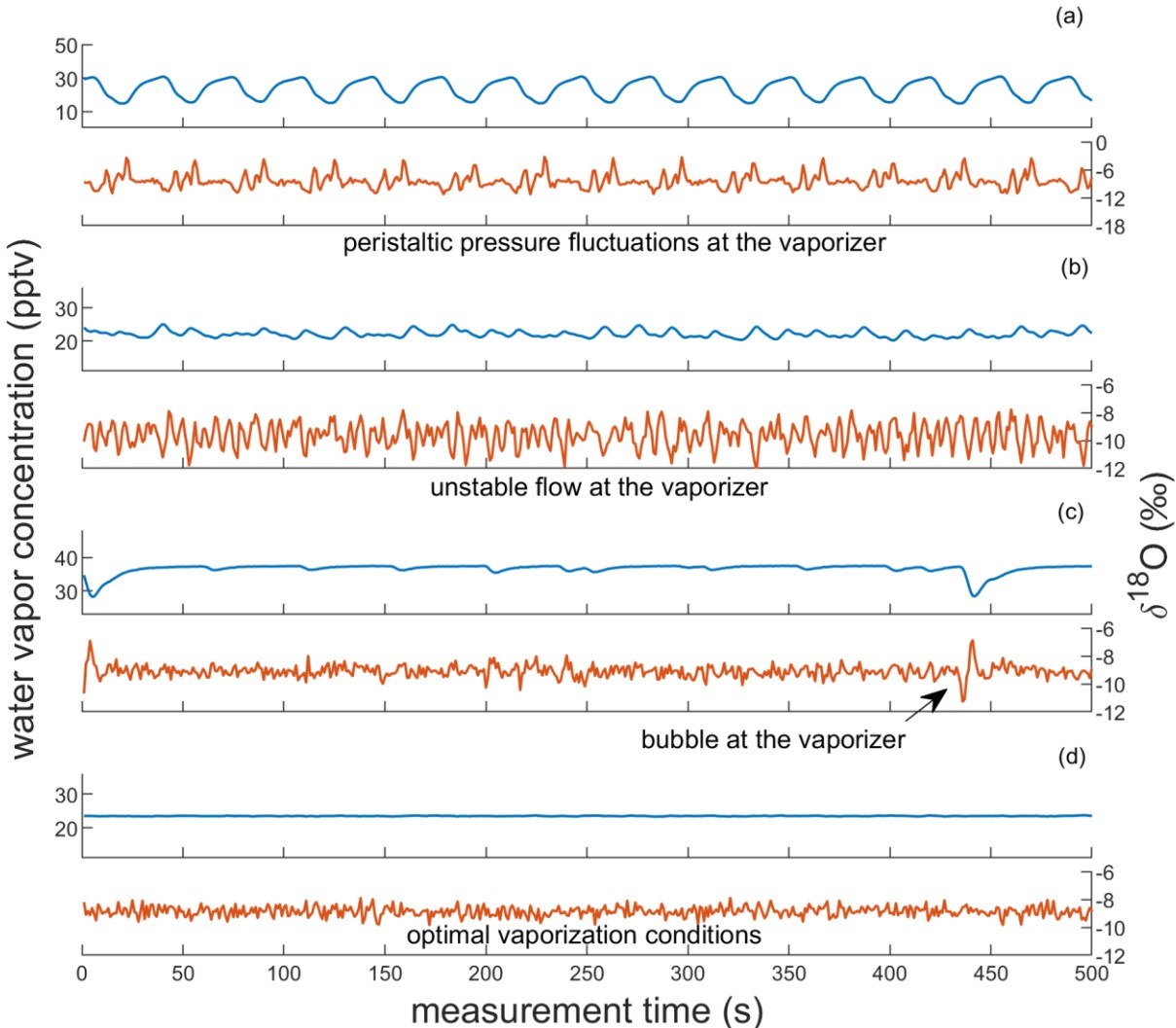

Figure 3: Observations of vapor quality as real-time indicators of vaporizer performance. Each subplot shows corresponding observations of water vapor concentration and $\delta^{18}O$ of SW2, reported parts per thousand vapor (pptv) and ‰, respectively. (a) and (b) show observations indicative of imbalanced vaporizer conditions for large and small pressure imbalances, respectively. (c) and (d) show observations indicative of acceptable vaporizer performance. Though both include low-variability observations of $\delta^{18}O$ ($\sigma$ <0.3 ‰) and of water vapor concentration, (c) also includes microbubble interruptions at the vaporizer (e.g., at 5 and 440 s). (d) indicates optimal vaporizer performance. Note that the vertical scaling of (a) is different from the other panels.

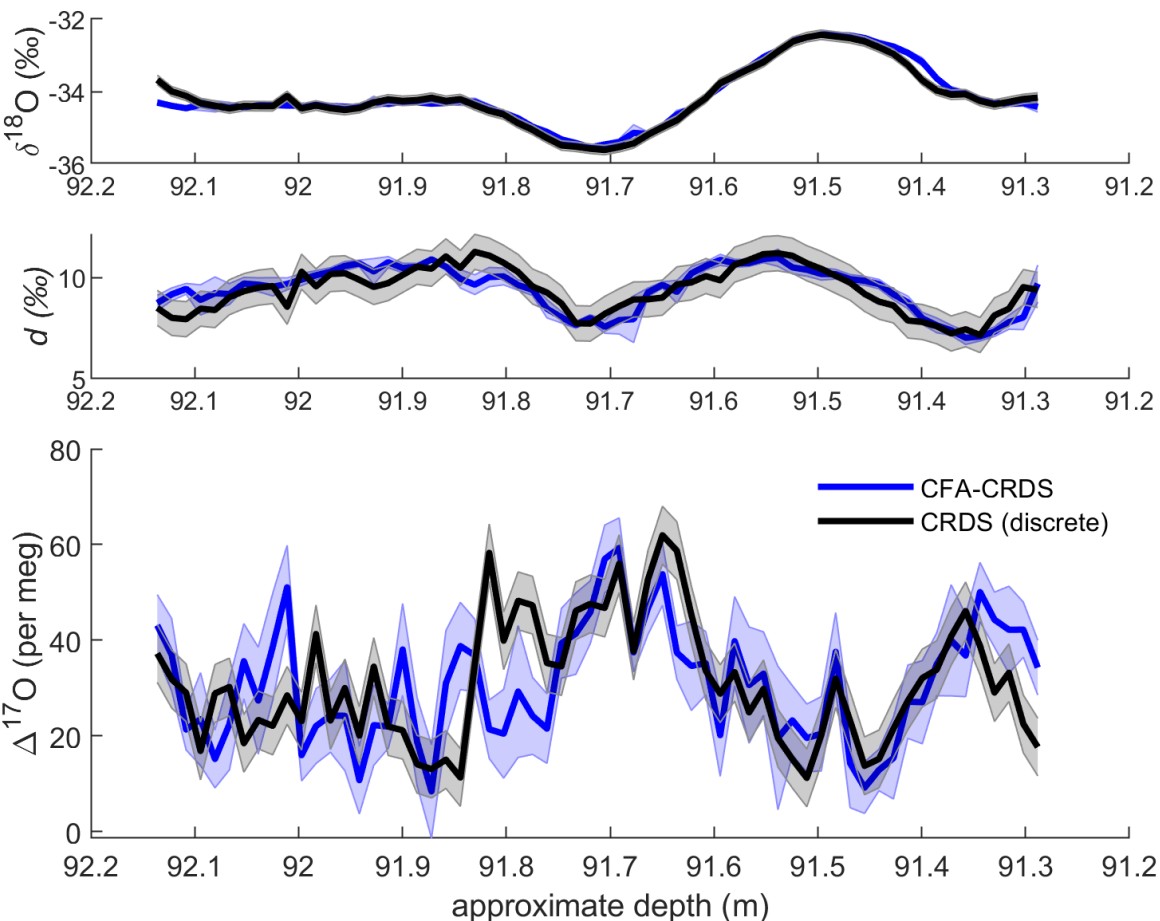

**Figure 4: Comparison of discrete CRDS ice core measurements (black) with calibrated CFA-CRDS data averaged over 1.39-cm intervals (blue). Corresponding δ18O and _d_ data are shown for seasonal context. Discrete CRDS measurements are shown with the root mean square error of corresponding reference water measurements (grey shading), and CFA-CRDS measurements are shown as the mean of nine measurements with the standard error (blue shading). Note that δ18O and _d_ are reported in ‰ and that Δ17O is measured in per meg; each vertical axis uses different scaling.**

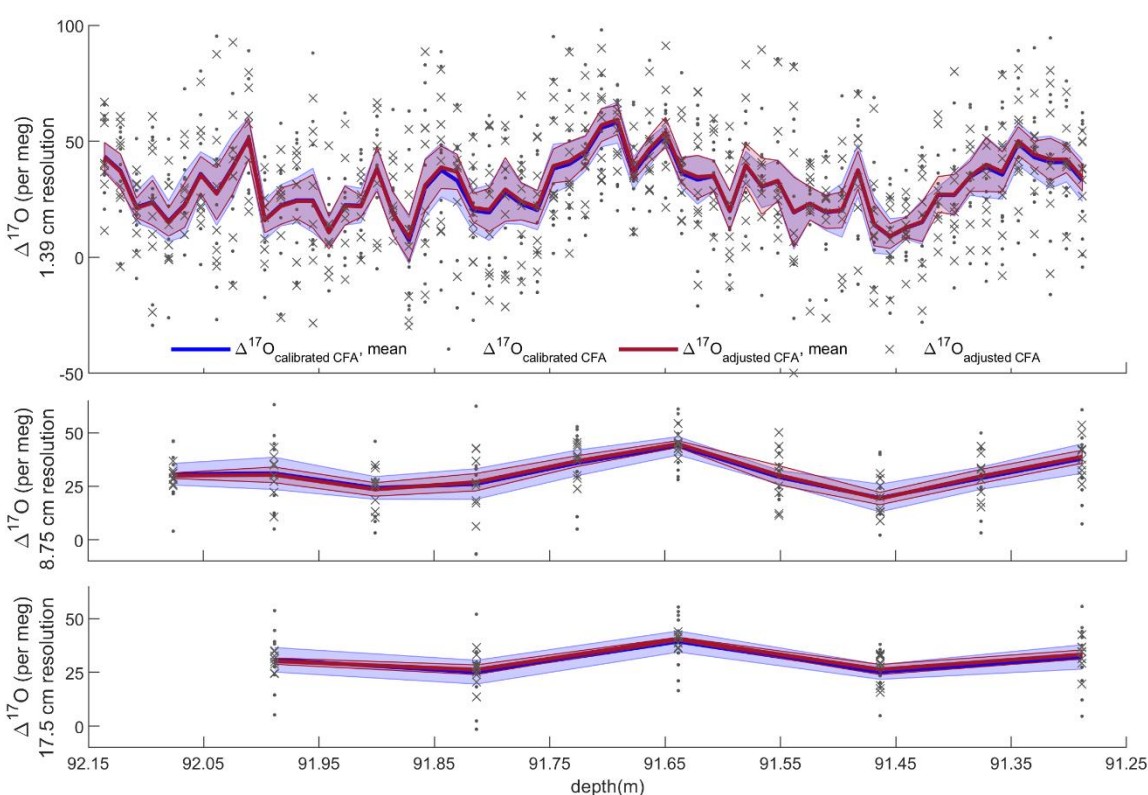

**Figure 5: Average Δ¹⁷O with standard error and all CFA-CRDS measurements, shown processed to three different depth resolutions.**
Dots and blue error envelopes indicate Δ¹⁷O$_{\text{calibrated CFA}}$, and x's and red error envelopes indicate Δ¹⁷O$_{\text{adjusted CFA}}$. All data are plotted at the upper depth of the depth interval that they represent. Note that the upper panel is expanded such that all three vertical scales are identical.

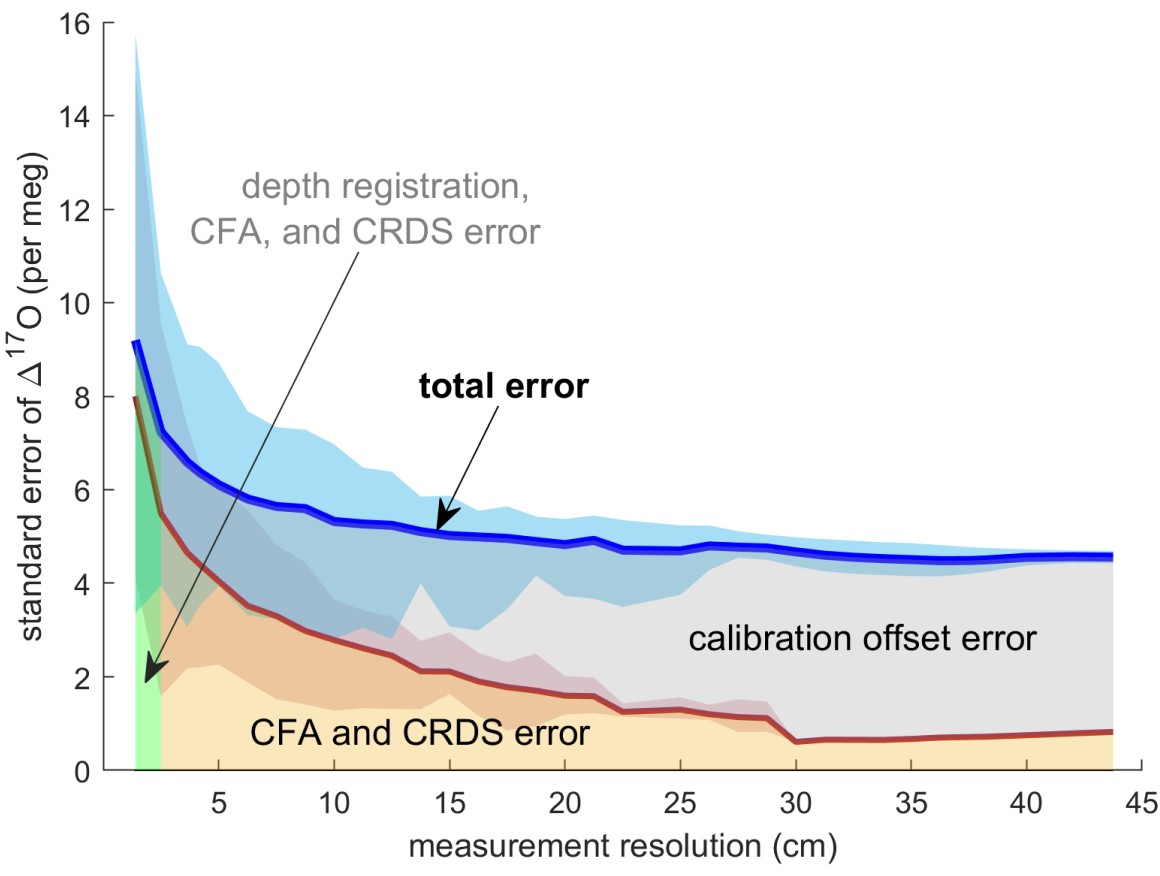

**Figure 6: Standard error of all replicate CFA measurements as a function of measurement integration depth. The blue line shows the mean of the standard error of $\Delta^{17}O_{\text{calibrated CFA}}$ as calculated for each depth interval; the shaded blue area indicates the minimum and maximum values of the standard error across all depth intervals. The red line and shaded area shows the relationship between the standard error in $\Delta^{17}O_{\text{adjusted CFA}}$ and the measurement resolution. The area beneath the total error line is highlighted to indicate**
**error attribution.**

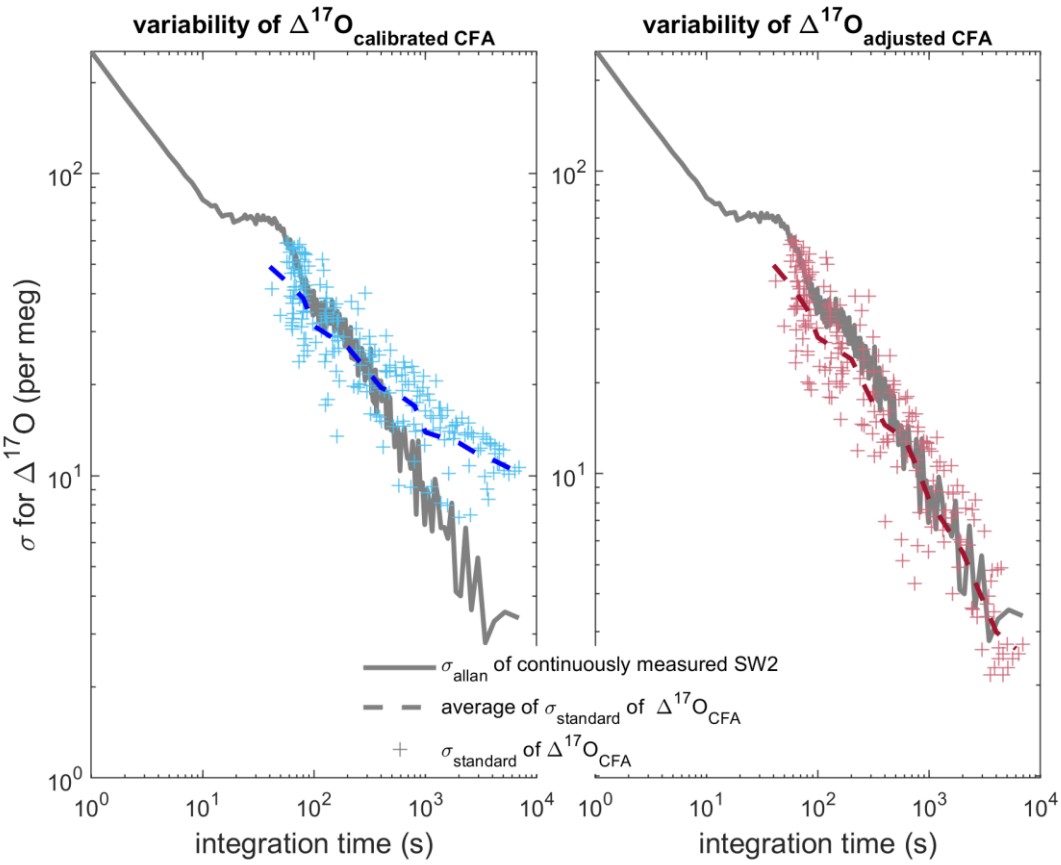

**Figure 7: Comparison of Allan deviation of continuous reference water measurements and standard deviation of nine duplicate CFA ice core measurements. In both left and right images, the Allan deviation line (solid grey) for a long measurement of SW2 is overlain by the standard deviation of the CFA-CRDS ice core measurements (crosses) and the mean of the standard deviations for each integration time (dashed line). The $\Delta^{17}O_{\text{calibrated CFA}}$ and $\Delta^{17}O_{\text{adjusted CFA}}$ data are shown in blue and red similarly to Figs. 5 and 6. The standard deviation on the left is calculated from calibrated replicate CFA-CRDS measurements and shows the total variability between CFA-CRDS replications along the depth of the core. The standard deviation information in the right plot is calculated from calibration-adjusted datasets so that the effect of the calibration offset error is removed; this analysis is still dependent upon instrumental noise, CFA errors, depth registration errors, and natural variability within the core.**

| Reference water (origin location) | $\delta^{17}O$ | $\delta^{18}O$ | $\delta D$ | $d$ | $\Delta^{17}O$ |
|---|---|---|---|---|---|
| | | ‰ vs. VSMOW | | | per meg vs. VSMOW |
| SW2 (Seattle) | -5.7107 | -10.85 | -77.96 | 8.84 | 33 |
| CW (West Antarctica) | -17.8807 | -33.64 | -265.95 | 3.17 | 25 |

| SPS2 (South Pole) | | -25.1210 | -47.07 | -365.20 | 11.36 | 15 |

Table 1: Isotopic values of reference waters. SW2 is Seattle deionized tap water; CW is melt water from the WDC06A core (i.e., West Antarctic Ice Sheet precipitation), and SPS2 is South Pole snow. These three waters were normalized to the VSMOW-SLAP scale using other in-house reference waters that were analyzed against VSMOW, SLAP, and GISP. The calibrated $\delta^{17}$O values are calculated from the combination of $\Delta^{17}$O and $\delta^{18}$O and are therefore reported to four significant digits (see Schoenemann et al., 2013 for additional details).

| JEMS2 (91.28-92.15 m) measurement date | SW2, CW, SPS2 measurement date | $m_{17}$ (unitless) | $m_{18}$ (unitless) | $b_{17}$ (‰) | $b_{18}$ (‰) | $\Delta^{17}$O offset (per meg) |
|---|---|---|---|---|---|---|
| 1) 01 Sept 2020 | 01-Sept-2020 | 1.0163 | 1.0068 | 4.5250 | -1.6133 | -13 |
| 2) 01 Sept 2020 | 02-Sept-2020 | 1.0094 | 0.9999 | 4.4551 | -1.6766 | -9 |
| 3) 02 Sept 2020 | 02-Sept-2020 | 1.0094 | 0.9999 | 4.4551 | -1.6766 | -6 |
| 4) 04 Sept 2020 | 04-Sept-2020 | 1.0089 | 0.9996 | 4.4396 | -1.6892 | 1 |
| 5) 08 Sept 2020 | 09-Sept-2020 | 1.0132 | 1.0040 | 4.5910 | -1.5211 | -25 |
| 6) 15 Sept 2020 | 15-Sept-2020 | 1.0065 | 0.9971 | 4.2982 | -1.9084 | 8 |
| 7) 25 Sept2020 | 25-Sept-2020 | 1.0058 | 0.9966 | 4.4130 | -1.6878 | 19 |
| 8) 09 Oct 2020 - 10 Oct 2020 | 10-Oct-2020 | 0.9983 | 0.9908 | 4.2542 | -1.8581 | 9 |
| 9) 10 Oct 2020 | 10-Oct-2020 | 1.0054 | 0.9967 | 4.4621 | -1.5888 | 5 |

Table 2: Sequence of CFA-CRDS measurements, including calibration information and calibration offset determined by Eq. (5) for 1.39 cm resolved data. Note that the long reference water measurement of SW2 used to generate Figure 7 was made on 18 September 2020. Also note that none of the reference water measurements on 8 September were acceptable to use for calibration, and that the large calibration offset for this measurement may be attributed to instrument drift or a change in CFA conditions between 8 and 9 September. Notable flow instabilities led to vaporizer cleanings on 14 September, and 8 October 2020, and no measurements were made between 27 September and 8 October 2020.

**Acknowledgements**

This work was partially funded by NSF award numbers 1841844 (Hercules Dome Ice Core) and 2019719 (Center for Oldest Ice Exploration). We are grateful to two anonymous reviewers for their thoughtful improvements to this manuscript. Finally, making continuous measurements during the COVID-19 pandemic would not have been possible without the support of University of Washington undergraduate students Jacob Childers and Shana Edouard, who assisted with system maintenance and measurements.

**Data Availability**

Data generated for this study are available from the corresponding author upon reasonable request.

**Author Contribution**

EJS, AJS, and LD conceived of the study. LD developed the measurement method, made the measurements, and completed the analysis with the support of AJS and EJS. All authors contributed to the manuscript.

**Competing Interests**

The authors declare that they have no conflict of interest.