# Peer review of "Improving continuous-flow analysis of triple oxygen isotopes in ice cores: insights from replicate measurements"

_EGUsphere, 2022_

## Author Comment (AC1)

Thank you for this detailed and instructive feedback. We have made several changes to the manuscript to address these comments; changes to the manuscript are noted below.

The abstract needs some adjustments to better summarize the main topic of this work, i.e. optimization of the CFA method, intercomparison of CFA-CRDS with discrete sample CRDS measurements, error analysis and noise contribution, etc. The authors should refrain to use strong wording such as "routine measurements", because if that would be the case then their current work would not be relevant. Also the claim of high-precision measurements of $\Delta 17O$ <5 per meg is questionable. Considering the reported 0.3 cm/min melt rate, the 1.4 cm resolution would correspond to about 270 s measurement time on CRDS that would result in about 25 per meg precision, in best case (see Allan deviation plot). The precision of ~5 per meg can eventually be achieved at averaging times longer than 3000 s, which corresponds to depth averages of ~15 cm. Obviously, more clarification is needed here.

We have updated the abstract to better reflect the conclusions of the paper, as suggested.

Another aspect that requires further discussion is the "calibration errors". First, it needs a clear definition and second, the interpretation of the data should be reconsidered. The authors present a large number (>40x, 3 h, over seven weeks) measurements of the reference waters (SW, SPS2, and CW), but they don't say/show anything related to the CW, although this is supposed to be used as an independent verification of the calibration. Also, the information about the two-point linear calibration curve (offset, slope and their variation across the individual measurements) is not given.

We have clarified the spread of the CW measurements in the manuscript and have added new text about the calibration error, including a new Fig. 4 that provides the calibration slope and intercept data.

Furthermore, a detailed description of the many limitation and critical role of a constant flow across the CFA is presented, but then the calibration gases are used on much lower flows compared to the ice core measurements. It is expected that changes in the flow would have a significant impact on the droplet formation in the vaporizer and that adsorption, memory effects, and isotope fractionation can also be changed. Considering all these details, it is difficult to see any direct and easy link that would allow for the conclusion drawn by the authors about the dominating role of "calibration error" in the CFA-CRDS method.

We have updated the text to clarify that downstream of the melt water vent tee, calibration standards and sample melt are treated identically, thus eliminating concern

about difference in fractionation within the vaporization system. The pressure at PI-1 is maintained at ambient conditions during both reference water and sample melt processing, and downstream flow rates are not adjusted.

Finally, the direct comparison of the CRDS stability (Allan variance) with the CFA reproducibility on Fig 7 is questionable. The Allan variance assumes a continuous stream of data, condition that is not fulfilled by the discrete measurements. Plots of the two-sample variance from different measurements based upon successive recordings of samples display a 'shoulder' effect of the reduced duty cycle on the system performance and the first deviation from the $1/\tau$ slope is a duty cycle effect and not an indicator for an accuracy (calibration) problem.

We believe that Reviewer #1 has misinterpreted our analysis. We do not use the Allan variance to evaluate calibration (i.e., accuracy) – rather, we use the Allan variance to demonstrate the precision of the system, and then compare it with the precision of CFA measurements. We have modified the manuscript text to make the analysis shown in Figure 7 more explicit.

Reviewer #1 suggests that two-sample variance from different measurements will display a "shoulder" effect of the reduced duty cycle, and we agree that the necessary lapse in time between our continuous measurements would affect the variance of the uncalibrated datastream due to, e.g., instrument drift occurring over long timescales between CRDS measurements. However, each of the nine CFA-CRDS timeseries used for this analysis has been calibrated to independent reference water measurements made immediately before the ice core data, and the variance among CFA-CRDS datasets is therefore a function of both the instrumental noise within the integration window of duration $\tau$ and also the calibration. The calibration should account for long-term drift in the mean value of the uncalibrated ice core measurements because the calibration standards would also be influenced by the drift. If the calibration is optimal, calibrated data should exhibit a similar relationship between variance and integration time as the Allan variance analysis. Despite this, we find that the precision of our calibrated data is worse than suggested by the Allan variance. We test the impact of the calibration intercept – which affects the magnitude of the mean signal but not the amplitude of the seasonal variability within the core. Shifting the calibration intercepts so that the mean values are equal results in a standard deviation to integration time relationship that mirrors the Allan variance relationship. This is similar to the intercept adjustment made by Steig et al. 2021 and suggests that calibration techniques can be improved with additional reference water information.

**Specific comments:**

Pg4, l.94. The authors should quantify the "several times more vapor".

We have clarified that the volume of water vapor produced at the vaporizer is approximately thirty times greater than required for analysis.

Pg4, l101. A pressure sensor monitors pressure and not flow conditions.

Agreed -- we have reworded this to be more precise.

Pg4, l111. Is there no issue (adsorption, memory) with having water carried over PFA tubing? Why do not use electro-polished stainless-steel tubing with inert coating instead?

Thank you for this suggestion. Although other materials may reduce adsorption throughout the system, transparent PFA tubing is advantageous for visually inspecting process lines while troubleshooting and was therefore selected for this work. We have added this detail to the manuscript.

Pg5, l130. The expression "analytical cavity" is not correct. I suggest using "optical cavity". Check for all instances across the manuscript. Also the "required sample volume" is not appropriate. The volume of the optical cavity is fixed as well as the pressure at which the CRDS operates.

We have updated this language as suggested.

Pg5, l131. What is meant by "measured volume"?

In some instances this had referred to the vapor that enters the optical cavity, and in others it referred to the liquid that enters the vaporizer; we rephrased this throughout for clarity.

Pg5, l134. Replace "all system instrumentation" by "sample handling system"

We have updated this language as suggested.

Pg5, l137. Consider simplifying the wording and replace "system instrumentation volume and system tubing diameter were minimized" with "the overall sample handling system volume was minimized"

We have updated this language as suggested for clarity.

Pg5, l140. Quantify the "excess liquid". How much compared to the measured sample?

Approximately 6x more water volume is vented and containerized than is vaporized. We have clarified this in the text.

Pg5, l143. Quantify "several times more"

We have clarified this in the text – approximately 6x more water volume and 30x more vapor volume is produced than is analyzed.

Pg.5, l145. The pump rate is minimized when measuring reference water. The authors should specify by how much is the flow reduced and comment on the expected effects due to the change in flow especially in terms of droplet formation in the vaporizer, adsorption, and isotope fractionation (see also my general comments).

While measuring reference waters, the PUMP-1 rate is reduced to match the flow rate of PUMP-2 (measured at 26uL/min). While measuring ice core melt, the PUMP-1 rate is set to accommodate the ice core melt stream (~450uL/min between sample and waste lines combined). The liquid line pressure at PI-1 and the flow rate setting at PUMP-2 are both held constant between sample and reference water measurements, so we do not expect or observe any flow changes at the vaporizer (the rate is ~26uL/min for both reference water and sample melt). We modified language about this in the manuscript for clarity.

Pg.7, l198. The anticorrelation between water vapor and δ18O is explained by two different effects: 1) incomplete vaporization, and 2) insufficient backpressure. The authors should be more consistent and clearly state which one applies. Is the correlation a real physical effect, due to e.g. isotope fractionation, mixing, etc., or is it simply reflecting the δ18O dependence on water vapor amount fraction of the CRDS?

We have reworded this section to clarify. We have observed that insufficient backpressure on the peristaltic pump causes larger, intermittent droplets to form within the vaporizer (instead of smaller, higher frequency droplets). These larger droplets do not completely vaporize within the vaporizer tee. The anticorrelation is a physical effect due to isotope fractionation within the vaporizer.

Pg.7, l201. What does the "apparent fractionation" means? An instrumental response?

We have updated this language to say "observed fractionation" – the vaporizer can be manipulated to cause fractionation within the chamber (by overwhelming the vaporizer with large droplets that do not vaporize immediately). The incomplete vaporization

response is characteristic and predictable. We have also pointed out this response with labels in Figure 3.

Pg.7, l205. The authors associate flow inconsistencies with isotope fractionation, but it is a misleading argument, because without excluding artifacts from the CRDS measurements itself it is difficult to disentangle the observed effects. In other words, the authors should discuss the mechanism behind the isotope fractionation generated by the variations in the flow.

We have reworded this text to better explain the relationship between the flow inconsistencies into the vaporizer and the observed fractionation.

Pg.7, l205. There is a list of many significant interventions: adjusting the peristaltic pump rate, replacing filter screens, adjusting FV-1, replacing peristaltic pump tubing, replacing or cleaning the capillary tube, or cleaning the vaporizer. The authors should give a more detailed discussion how often are these interventions necessary, what does it mean in terms of operation down-time and what is the impact on the calibration scale. Changing so many items should definitely result in rather different system response in terms of memory and surface effects, etc.

System changes are made before ice core measurements or between reference water measurements as needed. Filter screens were occasionally replaced during analysis, which caused brief flow disturbances at the vaporizer but did not otherwise impact analysis. This has been clarified in the text.

Pg.7, l210. The authors should explain how they are able to perform the automatic measurements while routinely tune the system to maintain steady pressure.

This language has been modified for clarity. We typically have considered the ongoing sequence of automated reference waters to be a "continuous" operation, but this is not the same as a continuous ice core measurement. The system is tuned before reference water measurements that are associated with ice core samples, or at other times in between reference water measurements when the system was observed to be out of balance. These adjustments were very intermittent because operators were rarely in the physical lab space aside from ice core measurements due to COVID-19.

Pg.8, l221. What is a manual observation?

We have clarified that this was a visual (and not electronic) observation of core height.

Pg.8, l244. What is the amount of rejected data compared to the entire set of measurements?

Due to COVID-19 lab restrictions, the reference waters were typically measured without oversight (reducing quality of some measurements if the system became imbalanced) and there were a few lengthy periods of downtime during the analysis window (when no operator was available for several days to reset the system). The measurements presented here represent approximately 50% of the analysis time. All measurements that were made with operator oversight are included in the final dataset.

Pg.9, l250. It would be very helpful to provide the scatter of the 47 individual 3 h measurements of the reference water to illustrate the stability of the CFA-CRDS system for this fundamental step.

We updated the text to state that the mean of all CW measurements is 25 and that the standard deviation is <12 per meg (n=53). For the most tightly clustered measurements of d18O, the mean of these CW measurements is 25 and the standard deviation is <6 per meg (n=36).

We also include here a comparison of CW variability for d18O and D17O to highlight the relationship between these calibrated datasets. To generate this figure, we took the standard deviation for d18O and D17O measurements of CW that had been filtered by progressively more restrictive variability thresholds for d18O. All measurements (n=53) have a standard deviation <0.25 per mil for d18O and <12 per meg for D17O, but we also know that some of these measurements were taken when vaporization conditions were not optimal (see Figure 3 for optimal conditions). Here we confirm that fractionation observed in d18O has a direct relationship to the errors in D17O – it also shows that all CW measurements are acceptably precise, but the best measurements

(as indicated by d18O variability) are also excellent for D17O (<6 per meg).

[Figure]

Pg.9, l264. This sentence is misleading. I suggest to modify it, e.g. "our ice core measurements cover of about two years period"

We have modified the text as suggested.

Pg.9, l273. In contrast to the author's statement, a correlation coefficient of 0.52 could either be interpreted as a "good" or "moderate" correlation, depending on the applied rule of thumb. The observed correlation may not necessarily be a good estimate for the population correlation coefficient, because samples are inevitably affected by chance. Therefore, the observed coefficient should always be accompanied by a confidence interval (95 %), which provides the range of plausible values of the coefficient in the population from which the data were sampled. Furthermore, the correlation coefficient of 0.52 corresponds to a coefficient of determination ($R^2$) of 0.27, suggesting that only about 27 % of the variability can be "explained". Finally, since both data are observations, a Pearson correlation analysis would be more appropriate here. In this case, both variables are assumed to be subject to natural random variation.

The Pearson correlation analysis was used to determine the correlation coefficient. The r value for the 1.4-cm data is significant with 99% confidence; this has been added to the text.

Pg.10, l274. Again, this statement is inappropriate since the deviations are comparable with the seasonal variation. It would be instructive considering a scatter plot using the CFA and discrete CRDS data shown in Fig7 (bottom).

Because the average CFA dataset includes nine measurements, the total analysis time associated with each depth interval is greater than 2000 s, and the noise expected (and observed) is substantially smaller than the seasonal variations. We have clarified this in the text, and include here the scatter plot that is suggested to show the correlation between CFA and discrete measurements.

[Figure]

Pg.10, l185. Although, this in principle holds, the slower ice melting would result in slow response time across the CFA, which would then have an impact on the achievable resolution. In general, instead of hypothesizing what would in principle be possible, the authors should consider only those cases that are realistic for high precision and routine ice core measurements.

We agree; we have removed this section from the text.

Pg.10, l289. The term of high-frequency instrumental noise is misleading. Use 1 Hz precision instead.

We have reworded this as suggested.

Pg.10, l292. The authors should explain what they mean under calibration error and what are the disproportionate drifts in d17O and d18O. Is there any systematic investigation of the oxygen isotope fractionation during vaporization? If yes, it would be

helpful for the reader to know its magnitude, reproducibility and dependency on various factors, such as flow, pressure, etc. Without these facts the claim cannot be proven.

We have reworded this section for clarity and have also added a new section and new Fig. 4, which clarify the calibration process and the effects of fractionation on the calibration values. We have also added text throughout the document to better describe calibration procedures and related errors.

In addition to Fig. 4, we below include a plot that examines changes in calibration error over time. Earlier work has identified that calibration offsets can impact the mean values of CFA-CRDS D17O data, so we are not surprised to find that over time, we observe similar calibration changes in our data. However, we do try to track these changes and use them to tailor maintenance and operating schedules, which we added to the discussion section of the manuscript. In the absence of fractionation, the calibration slopes and intercepts for d17O and d18O would be perfectly correlated. Monitoring the differences between calibration values for d17O and those for d18O therefore provides a way to directly assess changes in fractionation through time.

Here, we find the residual of m_d17O compared to the mean relationship between m_d17O and m_d18O, which has an R^2 value of 0.99 (blue line in the plot below is the residual of m_d17O). We do the same for values of b, which have an R^2 of 0.98 (orange line in the plot below). Despite very high correlations (and calibration differences that are completely inconsequential for d17O and d18O values), these very small deviations in d17O and d18O calibrations document fractionation that affects the calibration of D17O over time. In the plot below, the blue squares mark two times when the vaporizer fittings were cleaned. We have added some commentary about using these tools for method and calibration strategy development to the manuscript.

[Figure]

(y axis is in per mil for b (unitless for m), x shows the total duration of all measurements (same as Fig. 4)). Slope-derived residuals in blue, intercept-derived residuals in orange.

Pg.10, l315. The Allan variance analysis doesn't determine the theoretical variability, but the observed one.

We have clarified this wording as suggested.

Pg.10, l316. Define "internal noise".

We have corrected this to say "signal noise."

Pg.11, l320. This statement is not clear enough. If the reference water is handled in the same way as the melt water then how is it possible that its measurement does not account for the variability from the CFA system?

We agree that this statement was confusing as written; we have reworded this section for clarity. The Allan variance assesses the signal stability of a single continuous stream of data, but we are interested in how the precision during the measurement window compares with the precision of calibrated measurements on longer timescales. The repeat CFA-CRDS measurements allow us to compare these two things because we can assess the variability of individual measurements about the mean value at any given depth in our core.

Pg.12, l341. Again, the 1.39 cm resolution would correspond to 270 s measurement time leading to about 25 per meg precision (1σ) according to the Allan variance. Thus, the signal-to-noise on the seasonal cycle is less than 2.

Each depth interval includes 2000+ seconds of data and we have shown that the precision of this dataset it substantially smaller than the seasonal cycle. We have clarified this throughout the text where combined CFA-CRDS measurement data are discussed.

Fig.7: The Allan variance plot has a remarkable character. It seems that the CRDS stability is extraordinary as the deviation continues to decrease even after $10^4$ second (2.8 h). The authors should comment on why do they stop at this stage and consider the 2 per meg as best precision. It would be very interesting to see how long this continues. The authors should perform even longer reference water measurements to explore the limits where drifts start to dominate. At such instrumental stability, there is no need to make any calibrations for hours. This would imply that a 50 cm ice core could be easily measured in one run. Why this is not shown in this work?

We agree that the signal is remarkably stable on these timescales. In this paper, we include an Allan variance for an 8.5-h measurement period of Seattle water because this is the longest run of a single reference water that was made during the ice core measurement period, and it is therefore the most representative of the system configuration that was used for this analysis.

There is a "bump" at 10 s averaging time. Normally, this indicate a periodic oscillation in the system. Can the authors comment on the origin of this deviation from the white-noise at that time-scale?

We have observed this feature in all data from the instrument and are confident that it is an instrumental effect and not related to our sample handling system. It is documented in Schauer et al., 2014 and in Steig et al., 2021. While we do not fully understand the cause of this feature, we do not use data on such short timecales and therefore this feature does not affect our analysis.

Technical comments:

Abstract l.8-10: I recommend combining the two sentences and remove the "recent advances", because the CRDS technology for oxygen isotope measurements is almost 10 years old without any significant development since then. My suggestion: "... continuous-flow analysis (CFA) methods coupled to CRDS instrument allow for simultaneous measurements ...."

We have revised this statement as suggested.

I also suggest deleting the last sentence in the abstract.

We have removed the last sentence as suggested.

Pg2,l.30. Replace "routine" by "were demonstrated" since many of the cited references give a demonstration and show the feasibility of the technology rather than presenting routine ice core measurements.

We have reworded this as suggested.

Pg2, l.32. Replace "laser spectroscopy instrument" with "a laser spectrometer"

We have revised this as suggested throughout the manuscript.

Pg3, l.67. I suggest to write "CRDS spectrometer (L2140-I, Picarro Inc.)"

We have revised this as suggested.

Pg3, l.84. Remove space between value and degree, i.e. 200°C (look for other instances in the manuscript as well)

We have corrected this spacing here and throughout the manuscript.

Pg4, l.98. Replace "near-instantaneous analyzer output" by "CRDS values". The authors can assume that the reader is familiar with the CRDS and knows that this instruments report the results on a 1 Hz rate. I suggest removing all instances of "instantaneous".

We have revised this as suggested, here and in other instances throughout the manuscript.

Pg4, l102. I suggest replacing the very vague sentence "Instantaneous instrument output that reflects the internal vaporizer conditions and monitoring of the CFA line pressures provide information that is used..." by "These information is used..."

We have reworded this as suggested.

Pg4, l107. Replace the "x" by "×". For sake of simplicity write 30×30 mm$^2$

We have changed this as suggested.

Pg5, l140. Change "ice-core" to "ice core". Check for all instances across the manuscript.

We have updated this throughout the text.

---

## Author Comment (AC2)

Thank you for this detailed and instructive feedback. We have made several changes to the manuscript to address these comments; changes to the manuscript are noted below in our response.

The manuscript describes an analytical set-up to measure D17O f water y continuous flow analysis on ice core samples. The performances of the system are evaluated by comparing the results on parallel ice core barrels taken at the same depth. The authors also describe the different effects that influence the stability of the results and hence the final uncertainty on the D17O measurements. This technical paper is useful even if it is not clear how it can directly be applied to routine measurements of D17O by CFA because the measurement time would be very long and it seems that many adjustments (or cleaning) should be performed during the period of measurements.

I detail my comments below:

- l.71: You mention that diffusion and mixing should be reduced for D17O but it is not the case for all isotopes. Why is it more important for D17O ?

As noted in the text, the magnitude of variability of D17O is substantially smaller than that of other isotope quantities (in this ice, 45 per meg versus several per mil). Additionally, D17O (by definition) has a nonlinear response to changes in d17O and d18O. Both are reasons to attempt to prevent mixing within a system when measuring D17O. Mixing should be limited when possible to achieve the highest fidelity measurements for any isotope analysis. We have clarified that reducing mixing is important for all ice core measurement systems.

- Paragraph from l. 92: from this reading, it seems that the authors need to continuously adjust the system during measurements which makes it quite complicated and it is really difficult to understand what is done exactly. Could the authors be more precise on how they detect the problem and what action they take. Some concrete examples would be helpful. Also if these adjustments should be done continuously, how is it possible to make long runs with a good stability ?

Typically, system adjustments are made several hours in advance of measuring an ice core, so that the reference water measurements that are closest in time to the ice core analysis are measured under the same conditions. It is possible to make long runs with good stability for days to weeks, though the exact timing of interventions is dependent upon the performance of system components (filtration, debubbler) and the quality of the water passing through them (i.e. due to particulate loading or mineral precipitation within system components). We have clarified this in the text.

- What is the difference between the experimental set-up presented here and the one used in the previous study (Steig et al., 2021) ?

The system used in this study has not previously been utilized or published and was designed for this study. The system used by Steig et al., 2021 is the system whose details are published in Jones et al., 2017, as discussed in Section 2.1.

- l.137: is there any mixing linked to the melting of the ice on the melt head ?

Though it is likely that some amount of mixing occurs at the melt head, we do not observe any mixing effects at the averaging times or representative depth intervals used for this study.

- "Uemera" should rather be "Uemura" (several occurrences)

Thank you for noting this; we have corrected all instances of this name in our manuscript.

- l.175: It is not clear how the system components are cleaned ? Which components ? How is the cleaning done ? It seems that the cleaning occurs very often and I am wondering how this can be done without affecting the continuity of the measurements. It is very important that the authors explain there cleaning procedure and especially how it is done while the measurements are being performed continuously.

We have clarified this in the text; vaporizer fittings were cleaned four times during the measurement window with soap and water. Cleaning occurred between measurements of reference waters, not during or adjacent to ice core measurements.

- l.188: similarly, what is meant by "routinely clean" ? As this seems to be an essential aspect of the measurement technique, this should be detailed and explained to be able to understand and evaluate this technique.

As above, we have better explained this method and frequency in the text.

- l.210 and following paragraph: The sequence of measurements of reference waters is not given. When are these waters measured ? How is the measurement of these waters organized with respect to the measurement of the ice cores ? Do you measure the 3 waters every day ? Every week ? A table explaining the sequence of measurements (reference water, ice cores) over the 7 weeks should be given.

The three reference waters are measured repeatedly in a continuous sequence over the period of seven weeks, as described in Section 3.2. The only times that reference water measurements are not being made are when 1) an ice core is being measured or when 2) after a period of multiple days, the vaporizer operations destabilized and personnel were not in the laboratory due to COVID-19. This has been clarified in the text, and we have included new figure 5 to make this sequencing clear.

- l.221 and following paragraph: I understand that the authors did an alignment of the d18O seasonal cycles for this study to match parallel records but for real CFA measurements, when there is a need to have access to the real depth, this technique is not adapted and I do not see how you can avoid measuring the evolution of the height of the melting ice barrels. In this paragraph and in general in the manuscript, it should be made clear what is done specifically for this study with the aim to estimate uncertainty from compilation of record at the same depth (where absolute depth record is not needed) and what is done for a routine CFA measurement (where absolute depth record is needed).

We agree that routine measurements of unknown ice samples require automated depth registration and we recommend strategies similar to Jones et al. (2017) or Bigler et al. (2011). For this study, we use visual observations of melt rate during analysis and later assign high-resolution depth assignments based on the cross-correlation of d18O with the discrete d18O data (of known absolute depth). We estimate uncertainty based on the variance of depth data at inflection points in the timeseries, which we have added to the manuscript. This paper does not describe a routine CFA methodology, but here we reiterate the importance of electronic level sensing equipment to establish a high-resolution depth record for routine operations.

- Section 2.6: again the calibration sequence is not clear. A table is needed to explain what is done every day and over the 7 week period. What is exactly done during the 3 hour measurements of reference water ? Also the range of acceptable mixing ratios is very large (20,000 – 50,000 ppmv). Do you really observe so large variations in a short time ? over what time period is estimated the sigma_18O of 0.5 permil ?

We have revised the calibration section to clarify this process, and have added the complete sequence of measurements in Fig. 4. We clarify that system water vapor concentrations typically only fall below 20,000 ppm or above 50,000 ppm during maintenance downtime or when an operator was not available to restabilize the system during the COVID-19 pandemic. The screening criteria of 0.5 per mil is applied to each three-hour measurement window for the reference waters.

- Section 3.1: It would be very useful to see the raw data instead of only the average and envelopes of the records. This would help understanding the correlation (r=0.52) which I do actually not find as "high" as written by the authors. It would also help to understand the difference between the different sections of the core.

We have clarified that the r=0.52 correlation is significant with 99% confidence. We choose not to show the individual measurements at 1.39-cm resolution in the paper because, as expected, the noise at this resolution is too high for this to be useful (~25 per meg from the Allan variance). We include here a plot of all individual CFA measurements to demonstrate the observable seasonality within this population.

In this figure, we show the mean of all measurements and the standard error envelope, as shown in the manuscript. The grey dots are all discretized CFA measurements, each representing approximately 270 s of data.

[Figure]

- l.292: it is impossible to understand exactly what you mean by "disproportionate drift in d17O and d18O" since no number is given nor any example. What amplitude of the drift ? Over which period ? What should be done to avoid this ?

We have reworded this section for clarity and have added all calibration information to Fig. 4 to demonstrate the magnitude and timing of the calibration drift. We have added recommendations for establishing a calibration strategy to the discussion section.

- Paragraph starting from l. 303: again this is very difficult to follow if we do not see the raw data.

We have added a new section to more fully explain the calibration process, and have better explained the treatment of this data by relating it to the calibration process.

- l.326 and following paragraph: Please show the raw data and then the different treatments + explain exactly what you mean by calibration so that the reader can understand what was done. Additionnal figures showing the different step as well as the measurements (raw data) of reference waters used are needed to understand what you mean by calibration.

We have added a new Figure 5 and Equation 3 to visualize the calibration components and better explain the calibration process. We have also rewritten this section with more specific language to emphasize the impact of the calibration intercept. All CFA-CRDS data is provided in Figure 5.

- At the end, we expect clear recommendations on how to perform routine CFA measurements to achieve a good D17O signal. So we would need a recommended sequence of measurements and calibration with numbers given – a table is recommended. Also please explain how you suggest that calibration should be done. In the present state, the manuscript is not really useful for the reader who wants to repeat this set up.

Our work shows that D17O can be detected by CFA-CRDS, but that a robust calibration strategy and stable system operations are both important to achieving good data. We have added clearer recommendations to the discussion section for those who are interested in measuring D17O by CFA-CRDS, though we cannot recommend a specific formula for CFA methodology because, in practice, it will depend on the full suite of analyses that are desired. For example, while slower and/or repeated measurements are advantageous for D17O, for gas measurements like CH4 the opposite is the case (more rapid melt rates are better and duplicate measurements are generally not possible because of large volume requirements). This particular configuration performs well for D17O, but it is likely that other established methods for measuring water isotopes by CFA are also sufficient as long as the calibration is accounted for.

---

## Author Response (AR2)

Thank you to two anonymous reviewers and the editor for the detailed and helpful comments listed below in black. Our responses to all feedback are listed in red, below.

**First review:**

The author's response do not fully address some of the raised questions which makes it difficult to re-review all aspects of the revised manuscript. Therefore, only a few technical aspects are listed here: Please specify the resolution in ice core depth that corresponds to the 5 permeg precision stated in the abstract. (i.e. 5 permeg only at d >= 30 cm, according to Fig. 6) as well as the number of repeated measurements required.

We updated the abstract to specify that 5 per meg is attained only after 3000 s of analysis time. The depth resolution depends on the melt rate utilized, so we choose to report by analysis time instead.

Please consider improving the readability of the manuscript by more concise statements and by removing obsolete statements such as "..., it would be ideal to identify and eliminate sources of calibration error so that such an adjustment is not necessary".

We have made this statement and others throughout the text more concise for clarity.

Fig2: improve resolution

The resolution of the figure should be sufficient when taken from the .png and not from the .docx file.

Fig4: add labelling A, B, ... to the graph. Consider adding the D17O values to the figure.

We have removed figure 4 while attending to another reviewers comments; this information is now provided in Table 2.

Fig7: consider adding the SW2 as derived from Fig4, i.e. compare continuous with sequential (discrete) measurements of reference water.

We appreciate the suggestion but believe this would add no new information and would make the figure unnecessarily cluttered. The relationship between integration time and measurement precision for D17O for SW2 in Figure 7 is similar to previously published work using similar instruments (see e.g. Steig et al. 2014, Schauer et al. 2016, Steig et al. 2021) so this response seems to be characteristic of the instrument and not limited by this sample preparation method. Over long timescales, the continuous injection performs better than discrete injections due to the loss of information between injections and the propensity for instrumental drift over time; it is also a faster measurement because it eliminates downtime between discrete pulses into the commercial vaporizer.

In the conclusion specify the required time for CFA-CRDS measurements to obtain the cm-scale resolution.

We have added to the conclusion section that it still requires >1000 s to achieve meaningful CRDS data, and that it therefore requires >1000 s per cm to measure at the cm-scale.

**Second review:**

The authors did a good job in addressing the different comments. Still, I feel that some comments were not really addressed in a useful way for the reader.

1- Figure 4 does not really show the sequence of measurements. It should be possible to provide a table with date and depth/samples analysed so that we see when only standards were measured and when ice core samples have been analysed (with relevant standards for the calibration). If you really want to stick with Figure 4, focus should be done on periods when ice core barrels have been analysed (including also the standard measurements performed for calibration).

We have removed Figure 4 and added this information to the new Table 2 as suggested. Except where noted, the continuous sequence of reference waters was measured repeatedly between the ice core measurements as discussed in the text.

2- I am surprised by the answer concerning the cleaning procedure. Only a cleaning with water and soap of the fittings permit to remove problem of air and water flow restriction.

During the measurement study, we used only soap, water, and physical agitation to clean the vaporizer tee. We have since adopted a simpler procedure that includes soaking the tee in lactic acid, but we choose not to describe this in the manuscript since we did not use that procedure during the study period.

3- I understand that showing the 9 individual series will display a high noise and this is fine. Still, it would be very useful to show them (in appendix ?) because there is no way that CFA measurements can ever be done in 9 replicated barrels which hence questions the usefulness of this analysis. Do you think that measuring 2 or 3 replicate barrels may be enough to have a useful stacked D17O signal ? THis should be added somewhere in the discussion of the manuscript.

We do find that stacking measurements of Δ17O is advantageous both for increasing analysis time (and therefore improving precision and/or resolution) and for reducing the calibration error. We have added a section about this to the discussion. We have also included the spread of data for three averaging times in the new figure 5; figure 5 shows the spread of D17O_adjusted and D17O_calibrated so that the seasonality and total variability at each resolution are clear.

**Editor's comments:**

Comments to the author:
Dear Authors,

Thank you for submitting to AMT. Both reviewers agree that your revised manuscript provides significant science, but still judge that clarity (presentation and scientific) of the paper is only fair. Following the reviewers' recommendation, I hereby decide that the article can be published subject to

minor revisions, but require that the reviewers' comments must be taken into account in order to improve the clarity of the article. In addition to these suggestions, additional minor and technical issues aiming at improving the clarity of the paper are listed below. This concerns particularly the offset correction or mean-subtraction-method, which is difficult to understand. I therefore suggest adding an appendix on this subject. Please mention changes to the manuscript not only in the 'tracked version', but also in your comments on the reviewers' and editor's remarks. This will greatly facilitate the task of following up your corrections. Please also check style guidelines for AMT (https://www.atmospheric-measurement-techniques.net/submission.html#manuscriptcomposition). In particular, please follow the house standard not to hyphenate modifiers containing abbreviated units (e.g. "3-m stick" should be "3 m stick"). This also applies to the other side of the hyphenated term (e.g. "3 m long rope", not "3-m-long rope"). e.g., in lines 110, 125, 126, 127 and many more.)

Hyphens have been removed from modifiers containing abbreviated units. The calibration adjustment method has been formalized within the manuscript text.

l. 12. Please add the resolution required to reach the 5 per meg or give another uncertainty-resolution pair that you find more useful for the reader.

We updated this statement to indicate that 5 per meg precision is attainable after 3000 s of analysis time.

l. 116-117. I suggest to remove parentheses here

This text has been updated as suggested.

l. 155-157. Please check punctuation to enhance the logic of the phrase. I suggest replacing the first comma by a full stop and the semicolon by a comma sign.

This statement was rewritten for clarity.

l. 196-198. It seems that pulsating flow is causing the observed anti correlation and insufficient back pressure might cause/facilitate these pulsations.

This section has been reworded for clarity.

l. 202. '10s to 100s' may be confused with 10 s to 100 s. Therefore, the notation '10s per meg', etc. for 'several 10 per meg' should be avoided.

This has been rephrased to avoid confusion.

l. 261 etc. One referee has asked for pointing out the fraction of rejected data and your response has indicated that about 50 % of the analysis time corresponds to acceptable measurement conditions. It seems, however, that the corresponding number is missing in the changed manuscript.
This detail has been added to the text in line 277.

l. 330. replace 'suboptimal' by 'eventually biased'

We changed 'suboptimal' to 'biased' as suggested.

l. 343 and before. Until here, calibration issues may lead to biases in intercept and slope. It is only shown much later that intercept biases are likely the culprit. You should explain why discussing the slope is neglected in the paragraph beginning at l. 343.

Calibration issues might be attributed to the slope or intercept (and likely both), though a correction to just the intercept can account for the offset. It is important to consider the influence of all calibration information, so we have instead removed references to the calibration intercept, and have added text to clarify the possible causes of this error.

l. 343 ff and 630. Can you clearly demonstrate that the shift technique (removal of the CFA-CRDS mean value) removes only offset b effects and not m etc ? I suggest to formalize the procedure mathematically and present it in an appendix.

Thanks for this comment. We have formalized the procedure mathematically within the manuscript text, which has greatly simplified all references to the two data treatments throughout the manuscript. Though the shift can be achieved by changing only b for both delta values, the error itself is inherently caused by effects of both m and b (and specifically the relation between m or b for d17O and d18O), so we have reverted to calling this the "calibration offset error" and removed any attributions to the intercept alone.

l. 343-346. This paragraph is confusing and the description does not make clear how the data has been processed. One possible source of confusion is using the term intercept when talking about the δ17O and δ18O data, but the mean value removal likely concerns the Δ17O data. This should be stated more clearly.

We have clarified how these two things are related by defining the Δ17O calibration adjustment based only on D17O values and also by defining it similarly to how it has been treated by others (i.e. as a function of d17O and d18O). Adjusting the final value to an accepted value of Δ17O can be achieved by shifting Δ17O directly or by shifting both component values of δ17O and δ18O to their accepted values. Because the accepted values for D17O have corresponding accepted values of d18O and d17O, it does not matter how the offset is defined – nudging d17O and d18O to their accepted values will also correct D17O, but it is not necessary to involve d17O and d18O to make this adjustment. Therefore, we have defined the simpler mean adjustment on D17O only and use this definition in our discussion of the offset error.

l. 600 etc. Please use lowercase a, b, etc in the figure caption to comply with figure content. Understanding could be enhanced by indicating that scales in subfigures are not always the same.

All figure captions that include labeled panels have been updated to use lowercase letters, and we have noted when subpanel scales differ.

l. 608 etc. As before, please harmonize capitalization of labels between legend and figure.

All figure captions that include labeled panels have been updated to use lowercase letters.

l. 318 + 319. The statements on the correlation coefficient (r=0.52 with 99% confidence) or (r=0.70 with 99% confidence) are confusing. Since r has been estimated from the sample, it has been determined with an uncertainty and some range of r-values should be associated to the 99% confidence interval. But no such range is given -- or do r-values given with two significant digits mean that the range is smaller than ±0.005 ? The uncertainty of r is particularly interesting for assessing the significance of the difference in r between the upper 50 cm of the core and the rest of the data.

Thank you for pointing this out; we agree that this was confusing. We have updated these statistics with the range of r estimates from a resampling experiment and listed them at a 95% confidence interval in the text. Additionally, the figure below captures the full range of estimated r values and demonstrates that in the upper portion of the core where the seasonal signal is most prominent, the range of r values is substantially different from the lower section of the core where the D17O signal is relatively flat in both discrete and CFA datasets.

[Figure]

l. 396. After calibration, this should likely be accuracy and not precision.

We updated all language after the data is calibrated to refer to the accuracy, error, or variability instead of the precision.

l. 515 and Table 1. All instances of 'dxs' should be replaced by the symbol d for deuterium excess, both in the axis label and in the figure caption.

All instances of dxs have been replaced by the symbol d.

Table 1. It is very surprising that $\delta^{17}O$ is given with more significant digits than $\delta^{18}O$. Usually, measurements of $\delta^{18}O$ are more accurate and thus require more digits than corresponding $\delta^{17}O$ values. Note that a better than 3-digit precision is necessary to obtain a 1-digit precision in $\Delta^{17}$.

Our data have been normalized to the VSMOW-SLAP scale using reference waters that have been analyzed against VSMOW, SLAP, and GISP, as described by the table caption. Our data for $\delta^{17}O$ and $\delta^{18}O$ are presented to the number of decimal places suggested by Schoenemann et al., 2013 to accompany measurements of $\Delta^{17}O$.

Figure 4. The definition of m differs from that in equation (3). Please explain, or better, use m-1 in per meg as ordinate axis label.

The definition of m in Fig. 4 (now Table 2) is the same as in Eq. 3. m is the slope of the calibration equation and is therefore unitless (carries units of ‰/‰). We replaced Fig. 4 with Table 2, but the units of m were not changed.

l. 618 Figure 6. indicates that the calibration intercept error (CIE) increases with increasing measurement resolution. Is there a physical reason behind this ? Has this been discussed in the text ? Better understanding the behavior of the CIE probably allows to improve the technique further.

Thanks for this comment. We have added a section to the text to address the apparent change in calibration error with depth resolution. Definitionally, the calibration offset error is determined only by the mean value of the CFA measurement and its offset from the accepted value and therefore it should not depend on the depth resolution. The grey shading in Figure 6 indicates the full range of possibilities for the total error in the dataset and also for the total error in the calibration-adjusted dataset. Additionally, at very small averaging times, the calibration offset noise is essentially indistinguishable from the instrumental noise (see new figure 4), so accounting for the offset does not improve the data quality as notably as it does when other sources of error have been minimized.

l. 619. Replace 'by' by 'as a function of'.

Changed as recommended.

---

## Author Response (AR3)

Thank you for the instructive feedback. Please find the original editorial comments listed below in black, and our responses that note changes to the manuscript in blue.

According to the text (l 354) the mean value of all measurements is set equal to some value, corresponding to the notation used in eq (5), where an adjusted value (Δ_adj) is obtained from a measurement value (Δ_cal) corrected by the difference of two constants (or means). According to basic statistical rules, the mean of the adjusted value is different from the mean of the non-adjusted value, but the standard deviation of the two data must be the same. This is in contrast to Figure 7, where the standard deviation of the non-adjusted value is shown on the left-hand side and the standard deviation of the adjusted value is shown on the right-hand side.

Either there is a flaw in the argument, or more likely, the procedure is different from what I have inferred. In this case it is necessary that the procedure be explained in more detail.

We have updated the adjustment equation to use sigma notation, which clarifies that the mean value used for the calibration adjustment is a mean across all depths of a single measurement, whereas the standard deviation information presented in Fig. 7 provides the variability among depth-aligned measurements for all CFA measurements. That is, Fig. 7 shows the standard deviation among all 9 CFA measurements, which is a function of the integration time, and the variability across all depths of a single CFA measurement is a function of the seasonality of the analyte. The calibration adjustment does not alter the variability across all depths of a single CFA measurement.

As minor points, please also address the items listed here below:

l. 30: check notation in eq(3)

We have corrected the superscript and added the multiplication symbol to this equation.

l. 95: add et al to Jones

We have updated this as requested.

l. 113: use 30 mm x 30 mm to give correct dimension

We have updated this as requested.

l. 144: 'to the extent practical'. Please check grammar of this phrase

We have rephrased this to say "where possible" instead.

l. 301: give -> gives

We have updated this as requested.

l. 354: set the mean value of all measurements equal to a particular value, or ???
We updated this text to clarify that the mean value across the ~meter of ice is set to the mean value of the discrete measurements.

We have also updated the equation and the text to clarify that the mean correction is a single constant value that is applied to measurements at all depths along the core.

l. 458: delete 'per cm'
We have updated this as requested.

Table 1: Please follow Schoenemann et al (2013) who use 3 significant digits for both, $\delta 17O$ and $\delta 18O$ (see Table 1 of that paper). It does not make sense to give $\delta 18O$ with only two digits after the decimal sign and $\delta 17O$ with four, especially when the 'difference' ($\Delta 17O$) is given at the per meg level (corresponding to the 3rd decimal place in both, $\delta 17O$ and $\delta 18O$).

While Schoenemann et al (2013) do report standalone d18O values to three decimal places in the referenced table in order to avoid rounding error effects on the calculation of D17O, they/we recommended using four decimal places for d17O when d17O and d18O have been measured simultaneously. Independently, neither d17O nor d18O is more precise than two decimal places. However, when D17O is measured, the d17O value normalized to VSMOW-SLAP is calculated from the d18O and the

D17O. This is because the reference water values are also defined in terms of D17O, not d17O. Therefore, we report d17O to four decimal places, and d18O to two decimal places. This convention, described in detail on page 589 in the paragraph immediately below equation 11 in Schoenemann et al (2013), has been adopted by the IAEA. We therefore make no changes of significant digits in the table, but we have added a note to the table to make this convention clear. We also note that, as we report them, the d18O (with two decimal places) and the d17O (with four) provide the correct/measured values of D17O.

Table 2: Please check data entries. It seems that m and b values correspond to the SW2, CW and SPS2 measurements (entries 2 and 3 are identical, as are the dates of these measurements). However, this is not the case for entries 8) and 9), which, however, have also been obtained at the same date.

Entries 2 and 3 correctly indicate identical calibration values, as the first set of calibration standards measured on that date were closest in time to both CFA measurements from the previous day. Entries 8 and 9 also correctly indicate different calibration values, as the calibration standards measured closest in time to entry 8 were measured the morning after the measurement was taken; the standards measured closest in time to entry 9 were taken later that same day. The continuous looping of calibration standards between measurements typically yields two complete sets of calibration data per day. No changes have been made to the table.